# SRSA: Skill Retrieval and Adaptation for Robotic Assembly Tasks

**Yijie Guo[1], Bingjie Tang[2], Iretiayo Akinola[1], Dieter Fox[1,3], Abhishek Gupta[1,3] & Yashraj Narang[1]**
[1]NVIDIA Corporation, [2]University of Southern California, [3]University of Washington

## Abstract

Enabling robots to learn novel tasks in a data-efficient manner is a long-standing challenge. Common strategies involve carefully leveraging prior experiences, especially transition data collected on related tasks. Although much progress has been made for general pick-and-place manipulation, far fewer studies have investigated contact-rich assembly tasks, where precise control is essential. We introduce **SRSA** (**S**kill **R**etrieval and **S**kill **A**daptation), a novel framework designed to address this problem by utilizing a pre-existing skill library containing policies for diverse assembly tasks. The challenge lies in identifying which skill from the library is most relevant for fine-tuning on a new task. Our key hypothesis is that skills showing higher zero-shot success rates on a new task are better suited for rapid and effective fine-tuning on that task. To this end, we propose to predict the transfer success for all skills in the skill library on a novel task, and then use this prediction to guide the skill retrieval process. We establish a framework that jointly captures features of object geometry, physical dynamics, and expert actions to represent the tasks, allowing us to efficiently learn the transfer success predictor. Extensive experiments demonstrate that SRSA significantly outperforms the leading baseline. When retrieving and fine-tuning skills on unseen tasks, SRSA achieves a 19% relative improvement in success rate, exhibits 2.6x lower standard deviation across random seeds, and requires 2.4x fewer transition samples to reach a satisfactory success rate, compared to the baseline. In a continual learning setup, SRSA efficiently learns policies for new tasks and incorporates them into the skill library, enhancing future policy learning. Furthermore, policies trained with SRSA in simulation achieve a 90% mean success rate when deployed in the real world. Please visit our project webpage https://srsa2024.github.io/.

## 1 Introduction

Humans excel at solving new tasks with few demonstrations or trial-and-error interactions. In robot learning, a key challenge is to similarly enable robots to learn control policies from sensory input in a data-efficient manner. Achieving data-efficient learning is crucial for deploying robots in diverse real-world environments, such as the household and industry. A compelling approach to efficient policy learning is the development of a foundation model or generalist policy that spans multiple tasks, as the model or policy can offer long-term efficiency gains by providing a strong base for adaptation to novel tasks. Significant advancements have been made in manipulation tasks, particularly in visual pre-training (Parisi et al., 2022; Nair et al., 2022), multi-task policy learning (Shridhar et al., 2022; Goyal et al., 2024), and policy generalization (Jang et al., 2022; Ebert et al., 2021).

Despite this progress, efficiently solving new tasks in contact-rich environments, such as robotic assembly, remains underexplored. Robotic assembly plays a critical role in industries like automotive, aerospace, and electronics, but learning assembly policies is uniquely difficult. These tasks require contact-rich interactions with high levels of precision and accuracy, compounded by the physical complexity of the environments, part variability, and strict reliability standards. Much of the existing research focuses on training specialist (i.e., single-task) policies for individual assembly tasks (Spector & Di Castro, 2021; Spector et al., 2022; Tang et al., 2023). Building on the strengths of these specialist approaches, we propose a novel method for tackling new assembly tasks. Our approach leverages a skill library – a collection of diverse specialist policies and associated information (such as object geometry and task-relevant trajectories) for various assembly tasks. These policies and data, regardless of the training strategies or learning approaches used to develop them, can be harnessed to efficiently solve previously-unseen assembly challenges.

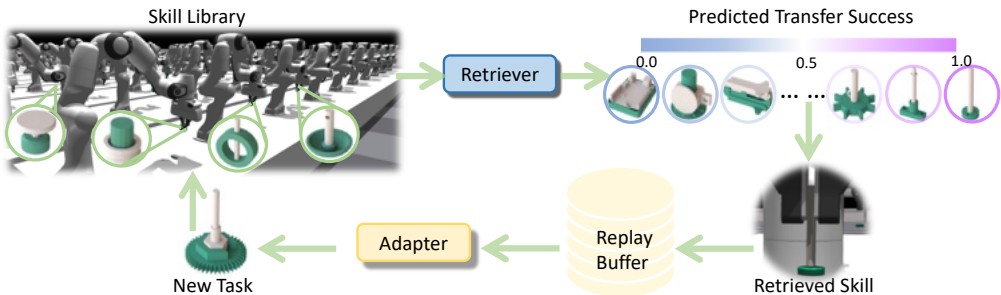

Figure 1: **Overview of SRSA.** We address assembly tasks, where the goal is to use a robot arm to insert diverse *plugs* (i.e., the white parts) into or onto corresponding *sockets* (i.e., the green parts). Specifically, we propose to predict the transfer success of applying prior skills (i.e., policies) to a new task, retrieve the skill with the highest predicted success rate, and fine-tune it on the new task. During fine-tuning, we accelerate and stabilize adaptation by incorporating imitation learning of high-rewarding transitions from the agent's own replay buffer.

To utilize prior task experiences, previous work on general pick-and-place tasks has explored methods such as imitating state-action pairs from expert demonstrations (Du et al., 2023; Lin et al., 2024; Kuang et al., 2024) and encoding sub-task skills as macro-action choices (Lynch et al., 2020; Pertsch et al., 2021; Nasiriany et al., 2022). Unlike these approaches, which focus on reusing data or sub-task skills, our approach centers on adapting *policies* from previous tasks to solve novel tasks. These policies encapsulate essential task-solving knowledge in a generative form, making them a valuable starting point for further refinement. Despite having access to a library of policies, identifying the most relevant ones for fine-tuning on new tasks is still an open question, and the success of fine-tuning hinges on making the right selection. In this paper, we introduce **SRSA** (**S**kill **R**etrieval and **S**kill **A**daptation), a novel framework designed to retrieve policies for similar tasks and adapt them to new tasks, as illustrated in Fig. 1. The key contributions of this paper are as follows:

**(1) Skill Retrieval Method**: We propose a skill retrieval method that simultaneously and explicitly learns embeddings for three fundamental components of assembly tasks: part geometry, interaction dynamics, and expert action choices. We subsequently introduce a novel objective that leverages these embeddings to predict transfer success between any source policy and target task, implicitly capturing additional critical factors for policy transfer. This approach enables the effective retrieval of relevant skills, resulting in higher zero-shot transfer success when applied to new tasks.

**(2) Skill Adaptation Method**: We propose a skill adaptation method that fine-tunes retrieved skills on new tasks while incorporating a self-imitation learning method (Oh et al., 2018) to enhance performance and stability during fine-tuning. In a simulation-based, dense-reward setting explored in the leading assembly baseline (Tang et al., 2024), SRSA achieves a relative improvement of 19% in success rate with 2.4x faster training and 2.6x lower standard deviation across random seeds. In simulation-based, sparse-reward settings without demonstrations or curricula (closely aligning with real-world fine-tuning scenarios), SRSA outperforms the baseline with a relative improvement of 135% in success rate. Furthermore, we demonstrate that policies fine-tuned in simulation can be directly transferred to real-world robots, achieving a 90% average success rate without the need for additional training. This capability of effectively fine-tuning policies in simulation on novel tasks, and transferring these policies to the real world in zero-shot, highlights the potential for deploying high-performance solutions in real-world assembly tasks.

**(3) Continual Learning with SRSA**: Instead of training numerous specialist (i.e., single-task) policies from scratch, we propose gradually expanding a small set of initial skills via retrieval and adaptation to cover a broader range of tasks. This strategy improves sample efficiency by over 80% compared to (Tang et al., 2024) and stays consistently efficient as the skill library and target tasks evolve. Thus, SRSA provides an efficient solution for accumulating a large-scale collection of skills.

## 2 RELATED WORK

**Robotic Assembly Tasks** Robotic assembly is a critical manufacturing process in the automotive, aerospace, electronics, and medical device industries, but *adaptive* robotic assembly (e.g., robustness to part types, initial part poses, perceptual noise, control error, and environmental perturbations) is largely unsolved. Research (Beltran-Hernandez et al., 2020; Luo et al., 2021; Narang et al., 2022; Tang et al., 2023; Zhang et al., 2023; Noseworthy et al., 2024) on adaptive assembly has seen significant growth in recent years. Despite advancements in datasets and real-world benchmarks for assembling small, realistic parts (Kimble et al., 2020; 2022; Willis et al., 2022; Tian et al.,

2022), the exploration of policy learning across a wide variety of parts remains relatively limited. Many recent efforts in robotic assembly have concentrated on perception (Fu et al., 2022; Wen et al., 2022) or planning (Tian et al., 2022; 2024), rather than learning policies that are robust to disturbances and noise. Additionally, the policy-learning efforts that have addressed the widest range of assemblies have typically been restricted to <30 parts (Spector & Di Castro, 2021; Spector et al., 2022; Zhao et al., 2022). The largest study, AutoMate (Tang et al., 2024), introduced a diverse dataset featuring 100 assembly tasks with simulation environments and 3D-printable parts, and explores policy learning across these tasks. However, its approach primarily focuses on learning specialist (i.e., single-task) policies *from scratch* without leveraging prior experience or knowledge from related tasks. In contrast, our goal is to solve novel assembly tasks by leveraging skills from previously-solved assembly tasks.

**Retrieval-based Policy Learning** Many studies have explored techniques for utilizing datasets from other tasks for pretraining, such as visual pretraining (Parisi et al., 2022; Nair et al., 2022; Xiao et al., 2022) and multi-task imitation learning (Jang et al., 2022; Ebert et al., 2021; Shridhar et al., 2022). Recently, in robotic manipulation, some works have investigated how to selectively incorporate of-fline data from other tasks during policy learning, i.e., retrieving prior data according to expert demonstrations on the target task (Nasiriany et al., 2022; Belkhale et al., 2024; Shao et al., 2021; Zha et al., 2024). For instance, Du et al. (2023) selects pertinent state-action pairs based on visual and action similarity from offline, unlabeled datasets and jointly trains a policy using a small amount of expert demonstrations and the queried data via imitation learning. Lin et al. (2024), on the other hand, emphasizes motion similarity rather than semantic similarity by retrieving state-action pairs based on optical flow representations, followed by few-shot imitation learning with expert demonstrations and the retrieved data. Kuang et al. (2024) takes a different approach by extracting a unified affordance representation from diverse data sources and hierarchically retrieving and transferring 2D affordance information based on language instructions to perform zero-shot robotic manipulation. These works primarily study *data retrieval* for general pick-and-place manipulation tasks. (Zhu et al., 2024) introduces a policy retriever for pick-and-place tasks, which selects policy candidates from a memory bank to align closely with the current input, based on the cosine similarity between instruction and observation features. In contrast to these works, we focus on challenging contact-rich manipulation tasks, especially investigating transfer success predictor for *policy retrieval*.

**Embedding Learning for Task and Skills** *Task* embedding learning has been extensively explored in meta-reinforcement learning and multi-task reinforcement learning problems, where shared knowledge across tasks can significantly enhance learning efficiency for new tasks. Most previous approaches focus on capturing task features related to visual appearance in 2D images or dynamics in transitions (James et al., 2018; Rakelly et al., 2019; Lee et al., 2020). Contrastive learning is often employed to bring similar tasks closer together in the embedding space while pushing dissimilar tasks farther apart (James et al., 2018). *Skill* embedding learning, on the other hand, leverages unstructured prior experiences (i.e., temporally extended actions that encapsulate useful behaviors) and repurposes them to solve downstream tasks. Existing methods typically train a high-level policy where the action space consists of the extracted skills (Pertsch et al., 2021; Nasiriany et al., 2022; Hausman et al., 2018; Sharma et al., 2019; Lynch et al., 2020). Although most previous approaches use skills to solve subtasks and combine sequences of skills for long-horizon tasks, we focus on selecting and adapting a single relevant skill for a new task; our tasks of interest are assembly tasks, which are typically short-horizon but difficult to train due to exploration challenges and precise control requirements. Additionally, we integrate multiple embedding-learning approaches by *jointly* capturing three fundamental components of assembly tasks: part geometry, interaction dynamics, and expert actions. We consolidate these perspectives for more robust task representation.

## 3 PROBLEM SETUP

In this work, we consider the problem setting of solving a new target task leveraging pre-existing skills from a skill library. This library contains policies, each designed to solve a specific previously-encountered task. Our approach is motivated by situations (Rusu et al., 2016; Tirinzoni et al., 2019; Huang et al., 2021) where an agent can draw on knowledge from previously-learned policies to adapt quickly to a new task at hand. Similar to the multi-task reinforcement learning (RL) formulation (Borsa et al., 2016; Sodhani et al., 2021; Calandriello et al., 2014), we consider a task space $\mathcal{T}$ where each task $T \in \mathcal{T}$ is defined as a Markov decision process (MDP) $(\mathcal{S}, \mathcal{A}, p, r, \gamma, \rho)$. In this formulation, $\mathcal{S}$ represents the state space, $\mathcal{A}$ the action space, $p(s_{t+1}|s_t, a_t)$ the transition dynamics, $r(s_t, a_t)$ the reward function, $\gamma \in [0, 1)$ the discount factor, and $\rho$ the initial state distribution.

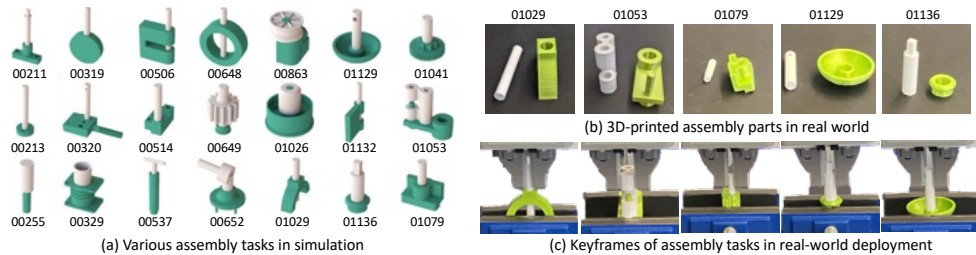

Figure 2: **Illustration of assembly tasks in AutoMate and SRSA**. (a) Samples of assembly tasks in the AutoMate benchmark. (b) 3D-printed parts of corresponding real-world assembly tasks in SRSA. (c) Keyframes from video recordings of our real-world deployments of performant policies.

Our study focuses on two-part assembly tasks, as depicted in Fig. 2. Following the setup of AutoMate (Tang et al., 2024), each environment includes a Franka robot, a *plug* (i.e., a part to be inserted), and a *socket* (i.e., the part that mates with the given plug). In the initial state, we randomize the robot's joint configuration and socket pose, as well as the pose of the plug within the robot's gripper. The goal of each task is to insert a plug into its corresponding socket. (See Appendix A.1)

The state space $\mathcal{S}$ consists of the robot arm's joint angles and velocities, the end-effector pose and its linear/angular velocities, the current plug pose, and the end-effector goal pose. The action space $\mathcal{A}$ consists of incremental pose targets for a task-space impedance controller. As described in (Tang et al., 2024), although assembly trajectories are infeasible to procedurally generate, *disassembly paths* can be easily generated, serving as reverse demonstrations that can be used by an RL agent. Specifically, the RL reward function is composed of terms that penalize the distance to the goal, penalize simulation error, reward task difficulty in a curriculum, and imitate the reversed disassembly paths. The assembly tasks all share the same state space $\mathcal{S}$ and action space $\mathcal{A}$, but vary in part geometries, transition dynamics $p$, and initial state distribution $\rho$.

Given a target task $T \in \mathcal{T}$, we assume access to a prior task set $\mathcal{T}_{prior} = \{T_1, T_2, \cdots, T_n\} \subseteq \mathcal{T}$. With policy space $\Pi : \mathcal{S} \rightarrow \mathcal{A}$, the *skill library* contains policies $\Pi_{prior} = \{\pi_1, \pi_2, \cdots, \pi_n\} \subseteq \Pi$ that solve each of the prior tasks, respectively. To solve a target task, the goal of RL is to find a policy $\pi(a_t|s_t)$ that produces an action for each state to maximize the expected return. We propose to first retrieve a skill (i.e., policy) for the most relevant prior task (Sec. 4.1), and then rapidly and effectively adapt to the target task by fine-tuning the retrieved skill (Sec. 4.2).

## 4 METHOD

### 4.1 SKILL RETRIEVAL

To effectively retrieve the skills from $\Pi_{prior}$ that are useful for a new target task $T$, we require a means to estimate the potential of applying a source policy $\pi_{src} \in \Pi_{prior}$ to the task $T$. Concretely, we aim to obtain a function $F : \Pi \times \mathcal{T} \rightarrow \mathbb{R}$, which takes as input a source policy and a target task, and produces a scalar score measuring how well the source policy can be adapted to the target task.

According to the simulation lemma (Agarwal et al., 2019), the difference in expected value when applying the same policy to different tasks partially depends on the difference in their transition dynamics and initial state distributions. We execute a source policy $\pi_{src}$ on both target task $T_{trg}$ and its original source task $T_{src}$. Let $r_{src,trg}$ denote the zero-shot transfer success of $\pi_{src}$ on $T_{trg}$ and $r_{src,src}$ its success rate on $T_{src}$. These success rates reflect the expected value of $\pi_{src}$ on $T_{trg}$ and $T_{src}$, respectively. Notably, if $r_{src,trg}$ is similar to $r_{src,src}$, it suggests that the transition dynamics and initial state distributions of two tasks may be closely aligned. Since $\pi_{src}$ is already an expert on $T_{src}$ with a high success rate $r_{src,src}$, a high zero-shot transfer success rate $r_{src,trg}$ indicates strong task similarity. Thus, we use the high transfer success rate as a heuristic indicator of similar dynamics and initial state distributions between source and target tasks. Details are in Appendix A.2.

Subsequently, we hypothesize that fine-tuning a source policy on a target task with similar dynamics will be efficient, as it only requires adaptation to small differences in dynamics. Therefore, we propose using zero-shot transfer success as a metric to gauge the potential to efficiently adapt a source policy to a target task. To identify a source policy with high zero-shot transfer success on a given target task, we propose to learn a function $F$ to predict zero-shot transfer success for any pair of source policy $\pi_{src}$ and target task $T_{trg}$. The prediction $F(\pi_{src}, T_{trg})$ serves as an indicator of whether $\pi_{src}$ is a strong candidate to initiate fine-tuning for the target task $T_{trg}$. Below, we describe data collection (Sec. 4.1.1), featurization (Sec. 4.1.2), training (Sec. 4.1.3) and inference (Sec. 4.1.4) for the transfer success predictor $F$.

Figure 3: **Illustration of skill retrieval approach.** We decompose skill retrieval into task feature learning(a-c) and transfer success prediction(d). (a) Geometry features are learned from point-cloud input using a PointNet autoencoder. (b) Dynamics features are learned from transition segments using a state-prediction objective. (c) Expert-action features are learned from transition segments using an action-reconstruction objective. (d) The zero-shot transfer success rate (of applying a source policy to a target task) is predicted using these task features from the source and target tasks.

### 4.1.1 DATASET FORMULATION

In order to train the prediction function $F$, we construct a dataset of tuples $(\pi_{src}, T_{trg}, r_{src,trg})$. We treat any two tasks from the prior task set $\mathcal{T}_{prior}$ as a source-target task pair. For each pair $(\pi_{src}, T_{trg})$, we evaluate the source policy $\pi_{src}$ on the target task $T_{trg}$ to obtain the zero-shot transfer success rate $r_{src,trg}$. In cases where multiple distinct policies exist for the same source task, each solving it in a different manner, policy-specific features would be necessary to capture nuances between different policies. However, in our setting, each policy in the skill library is trained as an expert for a specific source task, with a one-to-one mapping between policies and their corresponding training tasks. Consequently, we use the features of the source task $T_{src}$ as a proxy to represent the source policy $\pi_{src}$. This process enables us to collect a training dataset of tuples $(T_{src}, T_{trg}, r_{src,trg})$ from the prior skill library.

### 4.1.2 LEARNING TASK FEATURES

Given the limited number of $(T_{src}, T_{trg})$ pairs (specifically, during training, we have $n \times n$ pairs for a total of $n$ tasks in $\mathcal{T}_{prior}$), we need a strong featurization of both the source policy and target task for efficient learning of $F$. For assembly tasks, each task differs along three fundamental axes: part geometry, interaction dynamics, and expert actions that solve the task. Thus, we propose a framework that jointly captures features of geometry, dynamics, and expert actions to represent the tasks, allowing us to efficiently learn the transfer success predictor $F$ (Fig. 3).

When learning geometry features, we assume access to object meshes for both seen and novel tasks. This assumption is well-grounded in industry, where CAD models are widely available, allowing us to learn embeddings of 3D geometry. However, learning features for dynamics and expert actions poses a unique challenge. For new assembly tasks, we assume that expert demonstrations are *not* available, as these are typically tedious to obtain and often suboptimal for assembly tasks. This deficit prevents us from easily computing dynamics or action embeddings.

We draw insight from (Tian et al., 2022; Tang et al., 2024), which noted that, although procedurally generating assembly demonstrations for new tasks is intractable (narrow-passage problem), *disassembly paths* can be trivially generated by employing a compliant low-level controller to lift an inserted plug from its socket and move it to a randomized pose. We propose learning features for dynamics and expert actions using these *disassembly paths* and hypothesize that such features are useful for predicting transfer success for assembly. We later empirically support this hypothesis.

From each task, we randomly sample a certain number of points from parts' meshes as the point cloud $P$ and also randomly sample the transition segments $\tau$ from the disassembly trajectories. Using point clouds $P$ or transition sequences $\tau$, we learn encoders $E_G$, $E_D$, and $E_A$ to capture features $z_G$ (representing geometry), $z_D$ (representing forward dynamics) and $z_A$ (representing expert actions). We also train decoders $D_G$, $D_D$, and $D_A$ conditioned on these features to predict the point cloud for geometry, the next state for dynamics, and the action sequence for expert action choices. In Appendix A.4, we explain the implementation details for learning these features.

### 4.1.3 LEARNING TRANSFER SUCCESS PREDICTOR

We consolidate task features of source $T_{src}$ and target tasks $T_{trg}$ to develop the transfer success predictor $F$. We feed the sampled point cloud $P_{src}$, $P_{trg}$ and transition segments $\tau_{src}, \tau_{trg}$ from

$T_{src}$ and $T_{trg}$, into the pre-trained and frozen encoders $E_G$, $E_D$ and $E_A$. The geometry, dynamics, and expert action features are concatenated together to form task features $z_{src}$ and $z_{trg}$. We then pass the concatenated task features through an MLP to predict the transfer success $r_{src,trg}$, as illustrated in Fig. 3(d). Formally, we train the function $F$ to minimize the objective function (Eq. 1):

$$\mathcal{L} = \|F(\pi_{src}, T_{trg}) - r_{src,trg}\|^2 = \|MLP(z_{src}, z_{trg}) - r_{src,trg}\|^2$$
$$= \|MLP(E_G(P_{src}), E_D(\tau_{src}), E_A(\tau_{src}), E_G(P_{trg}), E_D(\tau_{trg}), E_A(\tau_{trg})) - r_{src,trg}\|^2 \quad (1)$$

### 4.1.4 INFERRING TRANSFER SUCCESS FOR RETRIEVAL

At test time, we use the well-trained function $F$ to predict the transfer success of applying any prior policy to a new task $T_{trg}$ as $F(\pi_{src}, T_{trg})$. As described in Sec. 4.1.2, for each task, we can randomly sample a certain number of points from parts' meshes as point clouds and randomly sample transition segments from the disassembly trajectories. For each pair of source and target tasks, we sample the input data for $m$ times and average the output of $F$ to obtain a more robust prediction of transfer success. Specifically, we sample point clouds $P_1, P_2, \cdots, P_m$ and transition segments $\tau_1, \tau_2, \cdots, \tau_m$, and then compute the averaged prediction for these samples, i.e., $F(\pi_{src}, T_{trg}) = \frac{1}{m} \sum_{i=1}^{m} MLP(E_G(P_{src,i}), E_D(\tau_{src,i}), E_A(\tau_{src,i}), E_G(P_{trg,i}), E_D(\tau_{trg,i}), E_A(\tau_{trg,i}))$. In this way, we infer the predicted transfer success $F(\pi_{src}, T_{trg})$ for any source policies $\pi_{src}$ in the prior skill library $\Pi_{prior} = \{\pi_1, \pi_2, \cdots, \pi_n\}$.

Although the well-trained $F$ provides transfer success prediction as an effective guidance for retrieval, its predictions may not always be perfectly accurate. To mitigate this, we retrieve the top-$k$ source skills ranked by the predictor $F$. Among these $k$ candidates, we identify the most relevant skill by evaluating their zero-shot transfer success on the target task, and ultimately select the skill with the best transfer performance. This technique is grounded in the same intuition as introduced in Sec. 4.1: zero-shot transfer success serves as a reliable metric for skill relevance. In the experiments in Sec. 5.2, we set $k$ to 5. Details are in Appendix A.12.

## 4.2 SKILL ADAPTATION

As mentioned in Sec. 3, our ultimate goal is to solve the new task as an RL problem. The retrieved skill is used to initialize the policy network $\pi_\theta(a_t|s_t)$, and we subsequently use proximal policy optimization (PPO) (Schulman et al., 2017) to fine-tune the policy on the target task. Our initialization provides a strong start for policy learning, as initial trials with the retrieved skills can achieve a reasonable success rate. Inspired by self-imitation learning (Oh et al., 2018), we fully exploit these positive experiences gained during the initial phase of fine-tuning. We maintain a replay buffer $\mathcal{D} = \{(s_t, a_t, R_t)\}$ to store the transitions encountered throughout the training, where $R_t = \sum_{k=t}^{T} \gamma^{k-t} r_k$ is the discounted sum of rewards. We prioritize the state-action pairs $(s_t, a_t)$ based on $R_t$ and imitate those pairs with high rewards. The objective function is defined in Eq. 2:

$$\mathcal{L}^{sil} = \mathbb{E}_{(s,a,R)\in\mathcal{D}}[\mathcal{L}^{sil}_{policy} + \beta \mathcal{L}^{sil}_{value}] \quad (2)$$

where $\mathcal{L}^{sil}_{policy} = -\log \pi_\theta(a|s)(R - V_\psi(s))_+$, $\mathcal{L}^{sil}_{value} = \frac{1}{2}\|(R - V_\psi(s))_+\|^2$, $(\cdot)_+ = \max(\cdot, 0)$, and $\pi_\theta$ and $V_\psi$ are the policy and value function (see the details in Appendix A.3).

As training progresses, the agent collects higher rewards on the target task, leading to an expanding replay buffer filled with improved experiences. As analyzed in (Tang, 2020), this self-imitation mechanism accelerates the agent's convergence to encountered high-reward behavior, even though it may introduce some bias into the policy. In our case, the behavior derived from the retrieved skill is advantageous for the target task. We find that self-imitation learning significantly enhances and stabilizes policy fine-tuning, proving especially beneficial in sparse-reward scenarios.

## 4.3 CONTINUAL LEARNING WITH SKILL-LIBRARY EXPANSION

Continual learning investigates learning various tasks in a sequential manner. The primary objective is to overcome the forgetting of previously learned tasks and to leverage earlier knowledge to achieve better performance and/or faster convergence on incoming tasks (Ring, 1994; Xu & Zhu, 2018; Abel et al., 2024). We integrate SRSA in the continual-learning setup and gradually expand the skill library. Specifically, we begin with an initial skill library $\Pi_{prior}$ corresponding to prior tasks $\mathcal{T}_{prior}$. When faced with a new batch of tasks $T^j = \{T_1, T_2, \cdots, T_k\}$, we apply SRSA to retrieve and fine-tune policies for each new task $T_i$. The learned policies are then incorporated as $\mathcal{T}_{prior} = \mathcal{T}_{prior} \cup \{T_i\}$; $\Pi_{prior} = \Pi_{prior} \cup \{\pi_i\}$. This approach allows us to efficiently tackle new

tasks by leveraging the skill library and simultaneously prevent the forgetting of all learned tasks by maintaining and expanding the skill library. See Appendix A.3 for the algorithm pseudocode.

## 5 EXPERIMENTS

We design experiments to answer questions: (1) Can SRSA retrieve source policies that achieve a better zero-shot transfer success rate on test tasks compared to baseline retrieval approaches? (2) Can SRSA policy fine-tuning improve learning performance, stability, and efficiency on test tasks? (3) After fine-tuning, can SRSA high-performing policies from simulation be deployed in zero-shot to the real world? (4) Can SRSA be applied in the continual-learning scenario to improve learning efficiency by gradually expanding a skill library? We investigate these questions on the AutoMate benchmark (Tang et al., 2024), which consists of 100 two-part assembly tasks with diverse parts, enabling us to study challenging contact-rich assembly tasks in simulation and the real world.

### 5.1 SKILL RETRIEVAL

AutoMate provides meshes and disassembly trajectories for each task. We use these data to learn the task embedding for retrieval. We compare the following retrieval strategies. **Signature**: retrieve the task with the closest path signature (Barcelos et al., 2024; Chen, 1958; Kidger et al., 2019), which represents the disassembly trajectories as a collection of path integrals (Tang et al., 2024). **Behavior**: retrieve the task with the closest VAE embedding of state-action pairs on disassembly trajectories. **Forward**: retrieve the task with the closest latent vector for the transition sequence on the disassembly trajectories, where the latent vector was trained to predict the forward dynamics. **Geometry**: retrieve the task with the closest PointNet (Qi et al., 2017; Wang et al., 2023) encoding for the point clouds of the assembly assets. **SRSA**: retrieve the source task with the highest prediction of transfer success on the target task. Implementation details can be found in Appendix A.4.

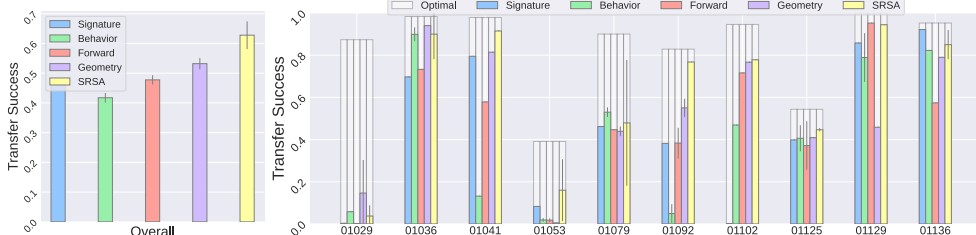

Figure 4: **Zero-shot transfer success of retrieved skills when applied to test tasks**. For each test task, we retrieve a policy from the prior skill library using 5 different approaches (4 baselines and SRSA). If the approach involves training neural networks, we train on 3 random seeds. *Optimal* represents the best transfer success rate on the target task among all source policies. **Left**: Mean and standard deviation of transfer success rate, averaged over 10 test tasks with 3 seeds each. **Right**: Mean and standard deviation of success rate for each test task, averaged over 3 seeds. Overall, SRSA substantially outperforms baselines.

Given the 100 tasks in the AutoMate benchmark, we split the task set into 90 prior tasks (to build the skill library) and 10 test tasks (as the new tasks to solve). For both SRSA and baseline methods, we train the retrieval model with three random seeds and report the average and standard deviation of transfer success across these seeds. Fig. 4 shows the result on the set of test tasks. SRSA performs best or second-best on all test tasks, except for one very challenging assembly, where all methods perform poorly (01029). In Appendix A.5, we show additional comparisons for other splits of prior and test task sets. Overall, SRSA retrieves source policies that obtain around 20% higher success rates on the test tasks, compared to baselines.

### 5.2 SKILL ADAPTATION

In this section, rather than investigating zero-shot transfer, we study policy learning on test tasks. We compare AutoMate learning specialist policies from scratch (Tang et al., 2024) and SRSA fine-tuning the retrieved specialist policy with self-imitation learning. Details are in Appendix A.4. We consider both the dense-reward setting (identical to AutoMate), which includes a reward term imitating disassembly demonstrations and a curriculum, and the sparse-reward setting, which only provides a nonzero reward for task success. The sparse-reward setting is designed to emulate real-world RL fine-tuning, where dense-reward information is much more challenging to acquire.

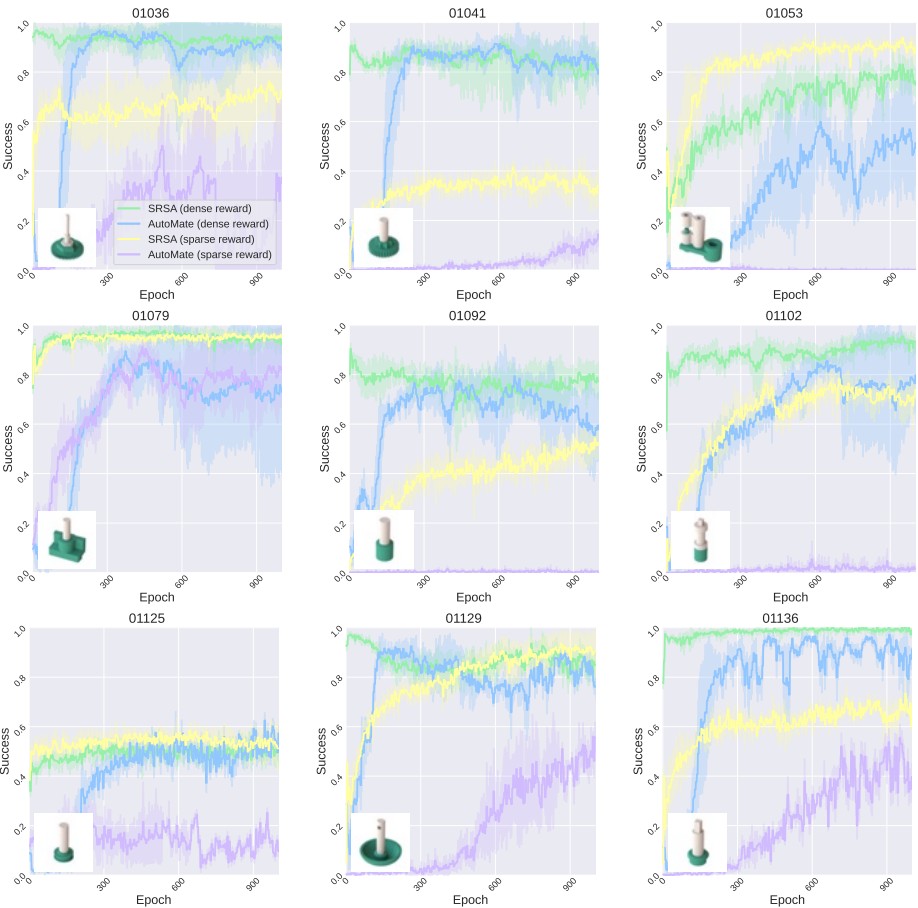

Figure 5: **Learning curves on test tasks.** The $x$-axis represents training epochs, where each epoch consists of 128 environment steps over 256 parallel environments. The $y$-axis represents success rate. The solid line shows the mean success rate over 5 runs with different random seeds, and the shaded area denotes the standard deviation.

Fig. 5 shows the learning curves in the set of test tasks. In the dense-reward setting, SRSA achieves strong performance with fewer training epochs than AutoMate. In the sparse-reward setting, AutoMate struggles to achieve a reasonable success rate, whereas SRSA benefits from the retrieved skill initialization and self-imitation learning, enabling it to reach higher performance. Additionally, in both settings, the learning curves of AutoMate exhibit instability with fluctuating success rates as the training goes on. Tab. 2 and Tab. 3 in Appendix A.5 summarize the mean and standard deviation of the success rate at the last epoch of training. These values are averaged across 5 random seeds for each test task. In the dense-reward setting, SRSA reaches an average success rate of 82.6% on 10 test tasks, outperforming AutoMate (69.4%), corresponding to a relative improvement of 19% in performance. Moreover, SRSA shows greater stability, as AutoMate exhibits a 2.6x higher standard deviation. In the sparse-reward setting, SRSA delivers a notable

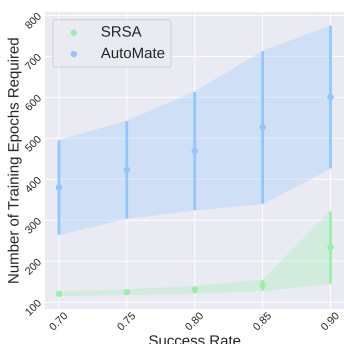

Figure 6: **Sample efficiency on test set**. To achieve a desired success rate (0.70, 0.75, 0.80, 0.85, or 0.90), we identify how many training epochs are required for each run. We illustrate the mean and standard deviation of required epochs across 5 runs with the points and error bars in the figure, averaged over 10 test tasks.

135% relative improvement in the average success rate compared to the baseline. Fig. 6 demonstrates the number of training epochs required to reach the desired success rate in the dense-reward setting. Averaged over 10 test tasks and 5 random seeds, SRSA requires far fewer training samples, i.e., at least 2.4 times fewer training epochs, to achieve an arbitrary success threshold.

5.3 REAL-WORLD DEPLOYMENT

We now deploy the trained specialist policies in the real world. As in ([Tang et al., 2024](#)), we place the robot in lead-through mode (a.k.a., manual guide mode), grasp a plug, guide it into the socket, and record the pose as a target pose. We then program-matically lift the plug until free from con-

| Asset ID | 01029 | 01053 | 01079 | 01129 | 01136 | Overall |
|----------|-------|-------|-------|-------|-------|---------|
| AutoMate | 7/10 | 1/10 | 7/10 | 4/10 | 8/10 | 54% |
| SRSA | 9/10 | 8/10 | 8/10 | 10/10 | 10/10 | 90% |

Figure 7: **Real-world evaluation.** We take the best checkpoint of policies across 5 runs within 500 epochs and report the success rate over 10 trials for each task.

tact, apply perturbations to the position and rotation of the end effector, and deploy a policy to assemble the plug into the socket. Such conditions emulate the perceptual noise and control error that are experienced in full robotic assembly pipelines. In Tab. 7, we take the best checkpoint in 500 training epochs in simulation and record its performance when deployed in the real world. In this relatively brief training time, SRSA reaches higher success rates than the baseline on real-world assembly tasks. We show keyframes of real-world deployments in Fig. 2(c). For videos, please refer to the project website https://srsa2024.github.io/.

## 5.4 CONTINUAL LEARNING

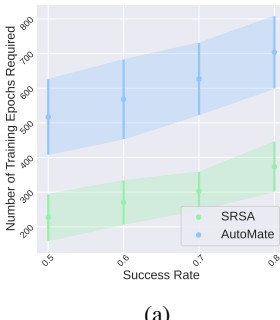
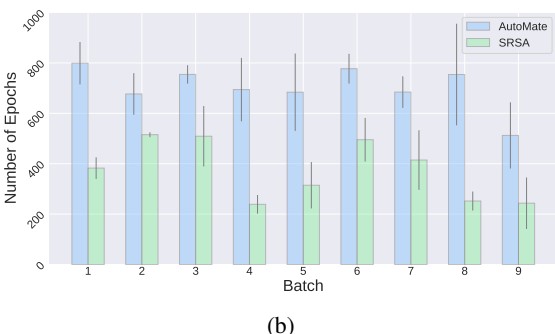

(a)                                                                        (b)

Figure 8: **(a) Overall sample efficiency**. We report the number of training epochs required to reach desired success rates (0.5, 0.6, 0.7, 0.8). We calculate the mean and standard deviation of the required training epochs over 5 runs, and report the average across 90 tasks. **(b) Sample efficiency in batches**. We sequentially introduce 9 batches of new tasks for policy learning, with each batch containing 10 new tasks. For each batch, we show the mean and standard deviation of training epochs required to reach a success rate of 0.8. SRSA consistently requires fewer training epochs.

We study the continual-learning setting to obtain policies for each of the 100 AutoMate tasks. Rather than training 100 policies from scratch in parallel, we start from a skill library with 10 tasks, and obtain 10 new policies for 10 new tasks utilizing the skill library. For each new task, we fine-tune the retrieved policy over 5 runs with different random seeds. We pick the best checkpoint with the highest success rate over 5 runs as the specialist policy for this new task. We repeat this process for 9 iterations, eventually covering the entire AutoMate benchmark. Essentially, we have a skill library that is gradually expanded with an increasing number of specialist policies.

In Fig. 8, we compare the sample efficiency of SRSA and AutoMate when learning specialist policies for the 90 tasks outside the initial skill library. We consider different desired success rates and report the number of training epochs required to reach each success rate. Overall, SRSA requires fewer training epochs to reach the desired success rate, demonstrating an 84% relative improvement in sample efficiency on average (Fig. 8(a)). For each batch of new tasks, SRSA is more efficient than the baseline, regardless of the skill library and test tasks (Fig. 8(b)). In Fig. 14, we show the success rates for the best checkpoints encountered in 5 runs for each task. SRSA achieves an average success rate of 79% compared to AutoMate's 70% across 100 tasks, while also exhibiting better training efficiency. In Appendix A.5, we present learning results for another ordering of batches of tasks, showing that the advantage of SRSA is insensitive to the order of encountering new tasks.

## 6 ABLATION STUDY

**Effect of Skill Retrieval** To verify the effect of skill retrieval, we conduct skill adaptation with retrieved skills using only a geometry embedding, i.e., the second-best skill retrieval approach eval-uated in Fig. 3. Fig. 9 shows the performance of policy fine-tuning for both SRSA and the geometry-based skill retrieval (SRSA-Geom). One can observe that retrieving a worse skill hinders learning

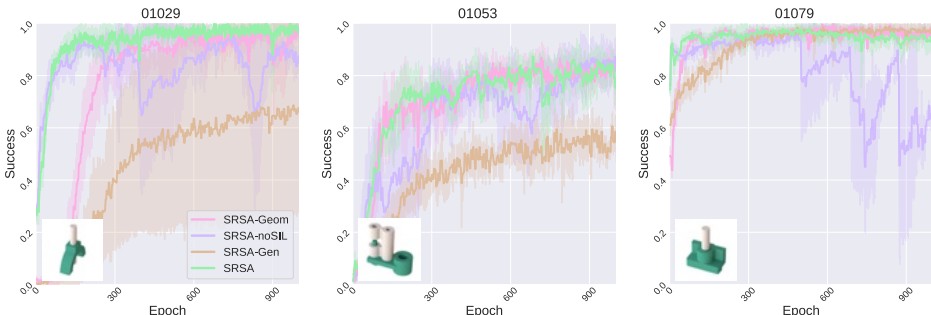

Figure 9: **Comparison across variants of SRSA**. For each method, we fine-tune the policy with 5 different random seeds. The learning curves show the mean and standard deviation of success rate over these seeds. We show learning curves for more tasks in the Appendix A.5.

efficiency, starting from a lower success rate and requiring more training epochs to reach high performance. Our retrieval approach SRSA improves adaptation efficiency over SRSA-Geom.

**Effect of Self-imitation Learning** To demonstrate the benefits of self-imitation learning (SIL) in policy fine-tuning, we compare SRSA to the variant without this component (SRSA-noSIL). In Fig. 9, SRSA outperforms the variant in terms of learning stability. In particular, SRSA-noSIL suffers from more fluctuations during fine-tuning and a larger standard deviation of success rate (shaded area) across runs with different seeds.

**Effect of Generalist Policy** We analyze whether fine-tuning a generalist policy outperforms fine-tuning a selected specialist policy. For policy initialization, we use the generalist policy for 20 training tasks from (Tang et al., 2024). Although it does not cover numerous tasks, it is the strongest generalist policy reported to date that can solve a diverse set of assembly tasks with an average success rate greater than 80%. Fig. 9 shows the learning curves of fine-tuning the generalist policy on unseen tasks (SRSA-Gen). We observe that SRSA-Gen provides a weaker initialization compared to SRSA, likely because the generalist policy's knowledge from the training tasks is less specialized than the skills retrieved by SRSA. Furthermore, adaptation is less efficient, possibly due to the larger neural network in the generalist policy, which requires more fine-tuning to adapt to new tasks. As a result, its asymptotic performance is also lower than that of SRSA.

## 7 CONCLUSION

**Summary**: In this work, we propose a pipeline to retrieve and adapt specialist policies to solve new assembly tasks. To learn a retrieval model, we jointly learn features from geometry, dynamics, and expert actions to represent tasks, and predict transfer success to implicitly capture other transfer-related factors from tasks. By combining skill retrieval with policy fine-tuning and self-imitation learning, our method efficiently learns high-performance simulation-based policies. We demonstrate that these policies are transferable to real-world robots. Additionally, we demonstrate that our approach can continuously expand a skill library through efficient learning of various skills.

**Limitations**: First, although we train policies for all assembly tasks in a leading benchmark (Tang et al., 2024), we do not address assemblies requiring rotational or helical motion (e.g., nut-and-bolt assembly). Second, we primarily concentrate on learning specialist (i.e., single-task) policies; future work could explore learning generalist (i.e., multi-task) policies, and, furthermore, incorporate knowledge from both specialist and generalist policies to solve novel tasks with even greater efficiency. Third, although our real-world success rates outperform the state-of-the-art in sim-to-real transfer for our examined tasks, they still fall short of the 99+% success rates required for industry-level deployment. We believe that RL fine-tuning directly in real-world settings could help bridge the sim-to-real gap and further improve success rates.

**Future Extensions**: How to utilize existing policies for new tasks (rather than training from scratch) is an open and general question in robotics. This question is relevant not just for insertion tasks, but also for general pick-and-place tasks, dexterous manipulation tasks, advanced assembly tasks, etc. Most robotics tasks are governed by geometry, dynamics, and behavior/action. We believe that our ideas of learning task features and predicting zero-shot transfer success for policy transfer can generalize to other domains. For instance, in tool-use tasks, the skill of using scissors may be beneficial for learning to operate pliers due to their similar shape and operating mechanism. We leave it as future work to extend SRSA to these additional robotics applications.

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

## A    APPENDIX

### A.1    ROBOT SETUP

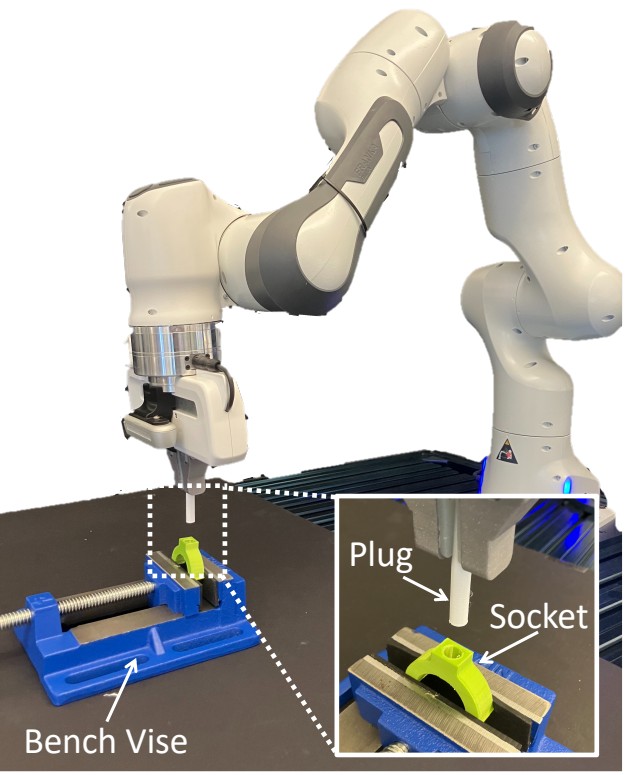

Figure 10: **Real-world experimental setup**. A Franka Panda robot and a bench vise are mounted to a tabletop. At the beginning of each episode, a 3D-printed plug is grasped by the robot gripper and a 3D-printed socket is haphazardly placed and clamped in the bench vise. The task is to control the robot arm to fully insert the plug into the socket.

### A.2    MOTIVATION WITH THEORETICAL PERSPECTIVE

Transferring knowledge from a source task to a target task can improve training efficiency and asymptotic performance. Consider a source task $T_j$ and target task $T_i$, which are MDPs that share state space $\mathcal{S}$, action space $\mathcal{A}$, and reward function $r$, but have distinct transition functions $p_i$, $p_j$ and initial state distributions $\rho_i$, $\rho_j$. To measure the transferability of a policy, we apply the same policy on both tasks and examine the difference in their expected values. Here we note that the value difference partially depends on the difference in their transition functions $p_i$, $p_j$ and initial state distributions $\rho_i$, $\rho_j$ (Proposition 1).

**Proposition 1.** *Let $T_i = \{\mathcal{S}, \mathcal{A}, p_i, r, \gamma, \rho_i\}$ and $T_j = \{\mathcal{S}, \mathcal{A}, p_j, r, \gamma, \rho_j\}$ be two MDPs in the task space $\mathcal{T}$. Applying a policy $\pi$ on $T_i$ and $T_j$, we have a function $f$ to describe the value difference:*

$$V^\pi(\rho_i, T_i) - V^\pi(\rho_j, T_j) = f(p_i - p_j, \rho_i - \rho_j)$$

*Proof.*

$$
\begin{aligned}
V^\pi(\rho_i, T_i) - V^\pi(\rho_j, T_j) &= \mathbb{E}_{s \sim \rho_i(\cdot)} \mathbb{E}_{a \sim \pi(\cdot|s)} Q^\pi(s, a, T_i) - \mathbb{E}_{s \sim \rho_j(\cdot)} \mathbb{E}_{a \sim \pi(\cdot|s)} Q^\pi(s, a, T_j) \\
&= \mathbb{E}_{s \sim \rho_i(\cdot)} \mathbb{E}_{a \sim \pi(\cdot|s)} [Q^\pi(s, a, T_i) - Q^\pi(s, a, T_j)] \\
&\quad + \mathbb{E}_{s \sim \rho_i(\cdot)} \mathbb{E}_{a \sim \pi(\cdot|s)} Q^\pi(s, a, T_j) - \mathbb{E}_{s \sim \rho_j(\cdot)} \mathbb{E}_{a \sim \pi(\cdot|s)} Q^\pi(s, a, T_j) \\
&= \mathbb{E}_{s \sim \rho_i(\cdot)} \mathbb{E}_{a \sim \pi(\cdot|s)} [Q^\pi(s, a, T_i) - Q^\pi(s, a, T_j)] \\
&\quad + \mathbb{E}_{s \sim \rho_i(\cdot)} V^\pi(s, T_j) - \mathbb{E}_{s \sim \rho_j(\cdot)} V^\pi(s, T_j) \\
&= \mathbb{E}_{s \sim \rho_i(\cdot)} \mathbb{E}_{a \sim \pi(\cdot|s)} [Q^\pi(s, a, T_i) - Q^\pi(s, a, T_j)] + \sum_s (\rho_i - \rho_j) V^\pi(s, T_j)
\end{aligned}
$$

For the Q-value difference $Q^\pi(s, a, T_i) - Q^\pi(s, a, T_j)$, we refer to the simulation lemma in (Agarwal et al., 2019):

$$
Q^\pi(T_i) - Q^\pi(T_j) = \gamma(I - \gamma P^\pi(T_j))^{-1}(p_i - p_j) V^\pi(T_i)
$$

where $P^\pi(T_j)$ denotes the transition matrix on state-action pairs induced by the policy $\pi$ on the task $T_j$, i.e., $P^\pi_{(s,a),(s',a')}(T_j) = p_j(s'|s, a)\pi(a'|s')$.

Consequently, $Q^\pi(s, a, T_i) - Q^\pi(s, a, T_j)$ is the $(s, a)$ item in the matrix $Q^\pi(T_i) - Q^\pi(T_j)$, and $Q^\pi(s, a, T_i) - Q^\pi(s, a, T_j)$ can be expressed as a function of $(p_i - p_j)$.

Thus, the value difference $V^\pi(\rho_i, T_i) - V^\pi(\rho_j, T_j)$ partially depends on $(p_i - p_j)$ and $(\rho_i - \rho_j)$. $\qquad \square$

Assume the reward function $r$ is a sparse, binary term indicating task success at the end of an episode. The success rate of applying a policy $\pi$ to a task $T$ can be represented as $V^\pi(\rho) = \mathbb{E}_{s_0 \sim \rho} \mathbb{E}_{\tau \sim p^\pi(\tau|s=s_0)} [\sum_{t=0}^\infty \gamma^t r_t]$. Here, our success rate $V^\pi(\rho_j, T_j)$ will naturally be high, as the source policy $\pi$ is already an expert policy for the source task $T_j$. When the success rate of applying the source policy to target task $T_i$ is also high (i.e., $V^\pi(\rho_i, T_i)$ is close to $V^\pi(\rho_j, T_j)$), then Proposition 1 implies that the transition functions $p_i$ and $p_j$, as well as the initial state distributions $\rho_i$ and $\rho_j$, might be similar. Consequently, if a source policy can achieve high zero-shot transfer success on a target task, the target task might have a similar transition function and initial state distribution as the source task. Hence, we hypothesize that fine-tuning the source policy on the target task will be efficient.

However, it is important to note that achieving a similarly high success rate on two tasks with a single policy does not necessarily indicate similar dynamics between the tasks. Proposition 1 establishes that similar dynamics and initial state distributions lead to similar expected values for a given policy, but the reverse is not guaranteed. We use the high transfer success rate as a heuristic indicator of similar dynamics, serving as intuitive motivation rather than strict theoretical justification.

## A.3 METHOD

---

**Algorithm 1** Policy Fine-tuning with Self-imitation Learning

---

Initialize parameters $\theta$ for policy $\pi_\theta$ and parameters $\psi$ for value function $V_\psi$ from retrieved skill
Initialize replay buffer $\mathcal{D} \leftarrow \emptyset$
Initialize episode buffer $\mathcal{E} \leftarrow \emptyset$
**for** each iteration **do**
    *# Collect training samples*
    **for** each step **do**
        Execute an action $a_t \sim \pi_\theta(a_t|s_t)$ in the environment and transit to the next state $s_{t+1}$
        Store transition $\mathcal{E} \leftarrow \mathcal{E} \cup \{(s_t, a_t, r_t)\}$
    **end for**
    **if** $s_{T+1}$ is terminal **then**
        *# Update replay buffer*
        Compute returns $R_t = \sum_k^T \gamma^{k-t} r_k$ for all $t$ in $\mathcal{E}$
        $\mathcal{D} \leftarrow \mathcal{D} \cup \{(s_t, a_t, R_t)\}$ for all $t$ in $\mathcal{E}$
    **end if**
    *# Update parameter using PPO objective with samples in $\mathcal{E}$*
    $\theta \leftarrow \theta - \eta \nabla_\theta \mathcal{L}^{ppo}$                                              (Schulman et al., 2017)
    $\psi \leftarrow \psi - \eta \nabla_\psi \mathcal{L}^{ppo}$
    *# Perform self-imitation learning*
    Sample a mini-batch $\{(s, a, R)\}$ from $\mathcal{D}$ according to advantages
    $\theta \leftarrow \theta - \eta \nabla_\theta \mathcal{L}^{sil}$                                                     (Equation 2)
    $\psi \leftarrow \psi - \eta \nabla_\psi \mathcal{L}^{sil}$
    Clear episode buffer $\mathcal{E} \leftarrow \emptyset$
**end for**

---

**Algorithm 2** Continual Learning with Skill Library Expansion

---

**Require:** Prior tasks $\mathcal{T}_{prior} = \{T_1, T_2, \cdots, T_n\}$; Skill library $\Pi_{prior} = \{\pi_1, \pi_2, \cdots, \pi_n\}$
1: **while** given new batch of tasks $\mathcal{T}^j = \{T_1^j, T_2^j, \cdots, T_m^j\}$ **do**
2:     **for** each task $T_i^j$ **do**
3:         Retrieve a policy $\pi_{src}$ from the skill library $\Pi_{prior}$
4:         Fine-tune $\pi_{src}$ to get a policy $\pi_k$ solving the task $T_i^j$
5:         Expand the skill library, $\mathcal{T}_{prior} = \mathcal{T}_{prior} \cup \{T_i^j\}$; $\Pi_{prior} = \Pi_{prior} \cup \{\pi_k\}$
6:     **end for**
7: **end while**

---

## A.4 IMPLEMENTATION DETAILS

### A.4.1 TASK FEATURE LEARNING IN SRSA

**Geometry Features**   As shown in Fig. 3(a), we employ a PointNet-based (Qi et al., 2017) autoencoder $E_G$ and $D_G$ to minimize the difference between input point cloud $P$ and reconstructed point cloud $D_G(E_G(P))$. The autoencoder is trained using point clouds of parts from all tasks.

We follow the implementation details outlined in (Tang et al., 2024). In a large set of meshes $M$ for various assembly parts, each mesh $m_i \in M$ consists of $(V_i, E_i)$, where $V$ denotes the vertices and $E$ represents the (undirected) edges. During each training iteration, we sample a batch of meshes $B \subset M$. For each $m_i \in B$, we generate a point cloud $P_i$ from the mesh, with each point located on the surface of $m_i$. The point cloud $P_i$ is passed through a PointNet encoder (Qi et al., 2017) based on the implementation from (Mu et al., 2021) to produce a latent vector. The latent vector $z_{G,i}$ is subsequently fed into a fully-convolutional decoder, following the implementation from (Wan et al., 2023) to produce the reconstructed point cloud $Q_i = D_G(E_G(P_i))$.

The network is trained to minimize reconstruction loss, defined here as the Chamfer distance between $P_i$ and $Q_i$:

$$\mathcal{L}_{CD} = \frac{1}{\|P_i\|} \sum_{p \in P_i} \min_{q \in Q_i} \|p - q\|_2^2 + \frac{1}{\|Q_i\|} \sum_{q \in Q_i} \min_{p \in P_i} \|p - q\|_2^2$$

Across 100 two-parts assembly tasks, we utilize a total of 200 meshes for the plug and socket components with $|M| = 200$. Each sampled point cloud $P_i$ contains 2000 points and the dimension of learned embedding is $|z_{G,i}| = 32$. The autoencoder is trained for a total of 23,000 epochs, using a batch size of 64 and a learning rate of 0.001.

To represent the features of one task, we gather the geometry features for the meshes of the plug, socket, and their assembled state, where the plug is fully inserted into the socket.

**Dynamics Features**   We build upon prior work in context-based meta-RL (Rakelly et al., 2019; Lee et al., 2020) to utilize a context encoder $E_D$ that produces a latent vector from transition segments $\tau_{t-1} = \{s_{t-h}, a_{t-h}, s_{t-h+1}, a_{t-h+1}, \cdots, s_{t-1}, a_{t-1}\}$, as shown in Fig. 3(b). We sample the transition segments from disassembly trajectories, compute the latent vector $E_D(\tau_{t-1})$, and feed the latent vector from transition segments to a forward dynamics model $D_D$ across all tasks. For any transition samples from any task, the forward dynamics model is trained to predict the next state $s'_{t+1} = D_D(E_D(\tau_{t-1}), s_t, a_t)$ to be close to the ground-truth next state $s_{t+1}$.

As described in (Tang et al., 2024), for each task, we generate disassembly paths by initializing the robot hand to grasp the plug in the assembled state, where the plug is fully inserted in the socket. Using a low-level controller, we lift the plug from the socket and move it to a randomized pose. We repeat this process until collecting 100 successful disassembly trajectories. We store the state and action at each timestep in the disassembly trajectories. Each task has a total of 100 disassembly trajectories, with each trajectory spanning 128 timesteps.

We sample the transition segment $\tau_{t-1} = \{s_{t-h}, a_{t-h}, s_{t-h+1}, a_{t-h+1}, \cdots, s_{t-1}, a_{t-1}\}$ for $h = 10$ timesteps. The context encoder is modeled as a multi-layer perceptron (MLP) with 3 hidden layers of sizes $(256, 128, 64)$, producing a 32-dimensional vector $z_{D,t} = E_D(\tau_{t-1})$. Then, the forward dynamics model $D_D$ receives the context vector as an additional input, where the input consists of a concatenation of state $s_t$, action $a_t$, and context vector $z_{D,t}$. The forward dynamics model comprises four fully-connected layers of sizes $(200, 200, 200, 200)$ with ReLU activation functions, outputting the prediction of the next state $s'_{t+1}$. The objective is to minimize L2-distance between the ground-truth next state $s_{t+1}$ and the predicted next state $s'_{t+1}$. For the entire set of disassembly trajectories across 100 tasks, we train the encoder and forward dynamics model for 200 epochs, using a batch size of 128 and a learning rate of 0.001.

**Expert Action Features**   We utilize the disassembly trajectories as reverse expert demonstrations for assembly tasks and aim to capture expert action information in an embedding space. As illustrated in Fig. 3(c), we sample a transition segment $\tau_t$ from the disassembly trajectories, map it to the action embedding $E_A(\tau_{t-1})$, and reconstruct the action sequence $\{a_{t-h}, a_{t-h+1}, \cdots, a_{t-1}\}$ using

decoder $D_A$. We train both the encoder and decoder with transition segments from all tasks. This embedding effectively extracts the strategy for solving the task by reconstructing the expert actions from the disassembly trajectories.

We sample the transition segment $\tau_{t-1} = \{s_{t-h}, a_{t-h}, s_{t-h+1}, a_{t-h+1}, \cdots, s_{t-1}, a_{t-1}\}$ for 10 timesteps (i.e., $h = 10$). The action encoder $E_A$ is modeled as a multi-layer perceptron (MLP) with three hidden layers of sizes $(256, 128, 64)$, producing a 32-dimensional vector $z_{A,t}$. The action decoder $D_A$ is an MLP with four hidden layers of sizes $(200, 200, 200, 200)$ that predicts the sequence of expert actions $\{a'_{t-h}, a'_{t-h+1}, \cdots, a'_{t-1}\}$. We minimize the L2-distance between input action sequence $\{a_{t-h}, a_{t-h+1}, \cdots, a_{t-1}\}$ and the reconstructed action sequence $\{a'_{t-h}, a'_{t-h+1}, \cdots, a'_{t-1}\}$. The encoder and decoder are trained for 200 epochs, using a batch size of 128 and a learning rate of 0.001.

### A.4.2 TRANSFER SUCCESS PREDICTION IN SRSA

We learn the function $F(\pi_{src}, T_{trg})$ to predict the transfer success. For any pair of source policy and target task in the skill library, we execute the source policy on the target task for 1000 episodes with randomized initial conditions and average the success rate to obtain the ground-truth label for $F$. For any task $T$ in the prior task set, we sample the point cloud $P_i$ of plug, socket, and their combined geometry (i.e. the plug is fully inserted in the socket) to extract geometry features $z_{G,i}$ with a dimension of 96. Then we sample transition segment $\tau_i$ to obtain the dynamics features $z_{D,i}$ with a dimension of 32 and action features $z_{A,i}$ with a dimension of 32. By concatenating these features, we create a task feature $z_i$ with a dimension of 160 for the sampled point clouds and transition segment. We feed the task features $z_{src,i}$ and $z_{trg,i}$ for the source and target tasks into an MLP with one hidden layer of size 128 to predict transfer success. We optimize the MLP to learn the transfer success prediction. The training is conducted for 50 epochs with batch size 64 across all source-target pairs in the prior task set.

### A.4.3 BASELINE APPROACHES FOR SKILL RETRIEVAL

**Signature**: Path signatures represent trajectories as a collection of path integrals and also quantify distances between trajectories. Inspired by (Tang et al., 2024), we find the closest path signature for skill retrieval. For each disassembly trajectory $\tau_k$ on the target task $T$, we calculate the path signature $z_k$ and search all disassembly trajectories over all source tasks to identify a source disassembly trajectory $\tau_j$ with the path signature $z_j$ closest to $z_k$. The source disassembly trajectory $\tau_j$ belongs to a source task in $\mathcal{T}_{prior}$; thus we match the target trajectory $\tau_k$ to this source task, denoted as $T_k$. We count the times that one source task $T_{src} \in \mathcal{T}_{prior}$ is assigned as the source task for a target disassembly trajectory, $C(T_{src}) = \sum_{k=1}^{n} [T_k = T_{src}]$. Then we retrieve the policy for the source task with the highest count, i.e., $\arg\max_{T_{src}} C(T_{src})$. In the case of ties, we select one at random.

**Behavior**: Inspired by (Du et al., 2023), we employ state-action pairs on disassembly trajectories across all tasks and learn a state-action embedding with a VAE for skill retrieval. For any state-action pair $(s_k, a_k)$ on the target task, we infer the embedding $z_{sa,k}$ and look for one state-action pair $(s_j, a_j)$ from the disassembly trajectories in source tasks with an embedding $z_{sa,j}$ closest to $z_{sa,k}$. The target state-action pair $(s_k, a_k)$ is matched to the one source task that $(s_j, a_j)$ belongs to. We denote this source task as $T_k$. Sweeping through all $N$ state-action pairs in the disassembly trajectories on target task, we count the times that one source task $T_{src} \in \mathcal{T}_{prior}$ is assigned as the source task for a target state-action pair, $C(T_{src}) = \sum_{k=1}^{N} [T_k = T_{src}]$. Then we retrieve the policy for the source task with the highest count, i.e., $\arg\max_{T_{src}} C(T_{src})$

**Forward**: We learn the latent vector for transition sequence $\tau$ on disassembly trajectories. In order to retrieve one source task according to the distances between task embeddings, we average the embeddings for all transition sequences from the same task to obtain the task embedding, similar to (Guo et al., 2022). We retrieve the policy for the source task with the closest task embedding.

**Geometry**: As explained above, we learn an autoencoder for the point clouds of the assembly assets to minimize the reconstruction loss, as conducted in (Tang et al., 2024). We retrieve the policy for the source task with the closest point-cloud embedding.

| Hyperparameters | Value |
|---|---|
| Policy Network Architecture | [256, 128, 64] |
| Value Function Architecture | [256, 128, 64] |
| LSTM network size | 256 |
| Horizon length (T) | 32 |
| Adam learning rate | 1e-4 |
| Discount factor ($\gamma$) | 0.99 |
| GAE parameter ($\lambda$) | 0.95 |
| Entropy coefficient | 0.0 |
| Critic coefficient | 2 |
| Minibatch size | 8192 |
| Minibatch epochs | 8 |
| Clipping parameter ($\epsilon$) | 0.2 |
| LSTM network size | 256 |
| SIL update per iteration | 1 |
| SIL batch size | 8192 |
| SIL loss weight | 1 |
| SIL value loss weight ($\beta$) | 0.01 |
| Replay buffer size | $10^5$ |
| Exponent for prioritization | 0.6 |

Table 1: Hyperparameters in PPO and Self-imitation Learning

### A.4.4 SKILL ADAPTATION IN SRSA

**Implementation Details** Following (Tang et al., 2024), we use PPO to train the stochastic policy $\pi_\theta$ (i.e., actor) and an approximation of the value function $V_\theta$ (i.e., critic), parameterized by neural networks with weights $\theta$. The policy is stochastic, following a multivariate normal distribution with a learned mean and standard deviation; however, at evaluation and deployment time, the action output from the policy is deterministic.

The input state for the policy network consists of the robot arm's joint angles, the end-effector pose, the goal end-effector pose, and the relative pose of the end effector to the goal. The state has a dimensionality of 24. Due to the asymmetric actor-critic strategy, the states provided to the value function include privileged information not available to the policy. The states for the critic include joint velocities, end-effector velocities, and the plug pose, resulting in an input dimensionality of 44 for the value function.

The action space consists of incremental pose targets, representing the position and orientation differences between the current pose and the target pose. We use incremental targets instead of absolute targets to restrict selection to a small, bounded spatial range. The action dimensionality is 6.

SRSA combines PPO with a self-imitation learning (Oh et al., 2018) mechanism for policy fine-tuning. We maintain a replay buffer $\mathcal{D}$ for transitions encountered during training. The data samples in the buffer are prioritized based on the discounted accumulated reward.

As shown in Algorithm 1, each iteration includes one PPO update for the policy and value function, along with a batch sampling from $\mathcal{D}$ to perform one self-imitation learning update. This update aims to minimize the loss function $\mathcal{L}_{sil}$ defined in Sec. 4.2. For details on network architectures and hyperparameters, refer to Tab. 1.

**Input Modality** We follow prior work (Tang et al., 2024) to use object poses rather than visual observations as input to the policy. Incorporating vision-based observations would introduce additional challenges for zero-shot sim-to-real transfer, as it requires a camera. In contrast, the current policy only relies on the fixed socket pose and the robot's proprioceptive features (including the end-effector pose), making it more straightforward to execute the policy in real-world settings.

Using visual observations or object pose is orthogonal to our proposed method (i.e., SRSA is independent of the observation modality). The high-level idea of retrieving a relevant skill and fine-tuning the retrieved policy remains applicable in scenarios involving vision-based policies. The geometry features derived from point clouds in our task representation can partially capture visual

similarities between tasks. This enables the retrieval of source tasks that are visually similar to the target task to some degree.

At the same time, SRSA may require modifications to better support vision-based policies, where each policy relies on a vision encoder to process high-dimensional visual observations. There is no guarantee that our retrieved source and target tasks are visually similar enough, and the features extracted by the vision encoder in the policy might differ significantly on the source and target tasks. This could pose challenges for fine-tuning the policy on the target task. To address this, we consider two distinct directions: 1. how to perform retrieval to better account for visual similarity; 2. how to train specialist policies with visual encoders such that the current SRSA retrieval strategy is still likely to work. Below, we propose specific approaches for these two directions.

1. Enhancing retrieval by incorporating features from visual observations: For example, integrating a Variational Autoencoder (VAE) to extract features from visual observations (as in BehaviorRetrieval (Du et al., 2023)) and combining these with other task representations might improve the retrieval process. Additionally, learning dynamics features, such as predicting future visual observations, could implicitly encode relevant visual information in task features for retrieval.

2. Improving the robustness of the visual encoder in the policy: Training the source policy with significant data augmentation (e.g., randomizing colors, poses, backgrounds, etc.) could make the visual encoder in the source policy more robust to diverse visual observations. It is more likely to extract similar features from geometrically similar tasks. Alternatively, leveraging state-of-the-art visual foundation models (e.g., DINOv2 (Oquab et al., 2023)) as visual encoders could further enhance generalization and robustness. These models have demonstrated strong performance in handling diverse observations and sim-to-real challenges, as shown in PoliFormer (Zeng et al., 2024). Consequently, we believe that features extracted by such visual encoders are likely to remain consistent for visual observations across geometrically similar tasks.

## A.5 EXPERIMENTS

### A.5.1 SKILL RETRIEVAL

We first replicate specialist policy learning for 100 assembly tasks as described in (Tang et al., 2024). These 100 tasks are then split into 90 prior tasks and 10 test tasks. For the 90 prior tasks, we use the trained specialist policies to build the skill library.

We train the task retriever on the prior tasks (Sec. 4.1) and evaluate its performance on the test tasks. In Fig. 4 in the main text and Fig. 11 and Fig. 12 in the Appendix, we present the test results for three different ways of splitting the 100 tasks. Overall, SRSA demonstrates superior performance in identifying relevant policies from the skill library, achieving a higher success rate in zero-shot transfer.

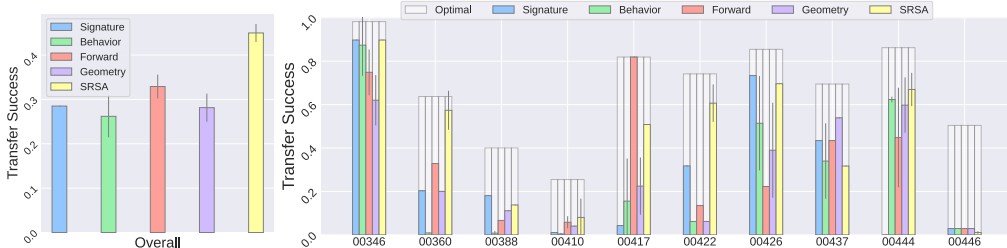

Figure 11: **Transfer success of retrieved skills applied to test tasks**. For each of the test tasks, we retrieve a policy from the prior skill library using 5 different approaches. For each approach, if it involves training neural networks, we train it for 3 random seeds. **Left**: we illustrate the mean result over 10 test tasks. **Right**: For each test task, we show the mean and standard deviation of transfer success over 3 seeds. Overall, SRSA clearly outperforms baselines.

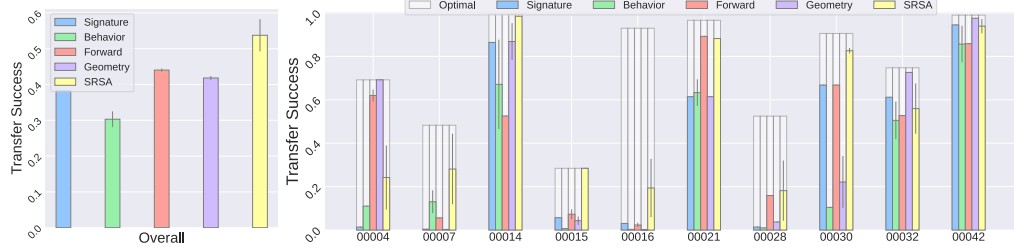

Figure 12: **Transfer success of retrieved skills applied to test tasks**. For each of the test tasks, we retrieve a policy from the prior skill library using 5 different approaches. For each approach, if it involves training neural networks, we train it for 3 random seeds. **Left**: we illustrate the mean result over 10 test tasks. **Right**: For each test task, we show the mean and standard deviation of transfer success over 3 seeds. Overall, SRSA clearly outperforms baselines.

### A.5.2 SKILL ADAPTATION

We show the learning curves in Fig. 5. At the end of 1000 training epochs, we record the success rate of the learned policies on 10 test tasks, as shown in Tab. 2 and Tab. 3. For AutoMate, the policies are learned from scratch using PPO. In contrast, SRSA initializes the policies with retrieved skills and fine-tunes them using PPO combined with self-imitation learning. The retrieval mechanism is trained on a skill library of 90 prior tasks, where the skills were pre-trained by AutoMate.

Compared to the baseline success rate of 69.4%, SRSA achieves a significantly higher success rate of 82.6%, corresponding to an absolute improvement of 13.2% and a relative improvement of approximately 19.0%. By leveraging the knowledge from the skill library, SRSA also obtains 2.6x lower standard deviation compared to AutoMate (Tab. 2). This advantage becomes even more pronounced in sparse-reward scenarios, where SRSA shows an absolute improvement of 40.8% and a relative improvement of 135% in comparison with the baseline. (Tab. 3).

| Task ID | 01029 | 01036 | 01041 | 01053 | 01079 | 01092 | 01102 | 01125 | 01129 | 01136 | Average |
|---|---|---|---|---|---|---|---|---|---|---|---|
| AutoMate | 53.4 (27.4) | 89.0 (7.7) | 79.1 (8.4) | 49.1 (15.3) | 74.3 (32.9) | 59.4 (13.1) | 76.4 (11.4) | 49.6 (3.2) | 76.0 (3.0) | 87.3 (4.2) | 69.4 (12.7) |
| SRSA | 97.3 (1.3) | 91.3 (6.0) | 78.9 (7.7) | 75.4 (6.4) | 90.5 (2.5) | 78.3 (6.3) | 86.6 (4.6) | 48.5 (5.7) | 82.3 (5.6) | 96.9 (2.0) | 82.6 (4.8) |

Table 2: **Mean (standard deviation) of success rate (%) on each test task in dense-reward setting**. We calculate the mean and standard deviation over 5 runs of different random seeds at the last training epoch (i.e., 1000 epochs).

| Task ID | 01029 | 01036 | 01041 | 01053 | 01079 | 01092 | 01102 | 01125 | 01129 | 01136 | Average |
|---|---|---|---|---|---|---|---|---|---|---|---|
| AutoMate | 61.3 (26.5) | 37.2 (31.4) | 14.4 (1.6) | 0 (0.5) | 81.7 (15.1) | 0 (0.5) | 1.4 (1.0) | 9.8 (2.0) | 55.6 (6.0) | 39.7 (5.4) | 30.1 (9.0) |
| SRSA | 95.1 (1.1) | 72.4 (8.9) | 33.7 (6.4) | 87.4 (3.6) | 96.1 (1.7) | 51.4 (5.5) | 70.7 (2.9) | 51.2 (9.3) | 90.3 (7.2) | 60.5 (2.6) | 70.9 (4.9) |

Table 3: **Mean (standard deviation) of success rate (%) on each test task in sparse-reward setting**. We calculate the mean and standard deviation over 5 runs of different random seeds at the last training epoch (i.e., 1000 epochs).

### A.5.3 CONTINUAL LEARNING

We begin with an initial skill library containing 10 policies and expand its size by 10 policies per round over 9 rounds, eventually reaching 100 policies. When the skill library contains fewer than 40 policies, the number of source-target task pairs from the prior task set is limited. During this phase, we retrieve skills solely based on geometry embeddings. That is to say, the retrieved skill

from the skill library is the one with the closest geometry embedding to the new task. Once the skill library reaches 40 or more policies, we train the transfer success prediction function $F$ to guide skill retrieval for new tasks.

In the continual-learning setting, Fig. 8 in main text and Fig. 13 in Appendix show the efficiency of SRSA and AutoMate under two different task batch orderings. In both cases, SRSA demonstrates significantly better sample efficiency compared to AutoMate.

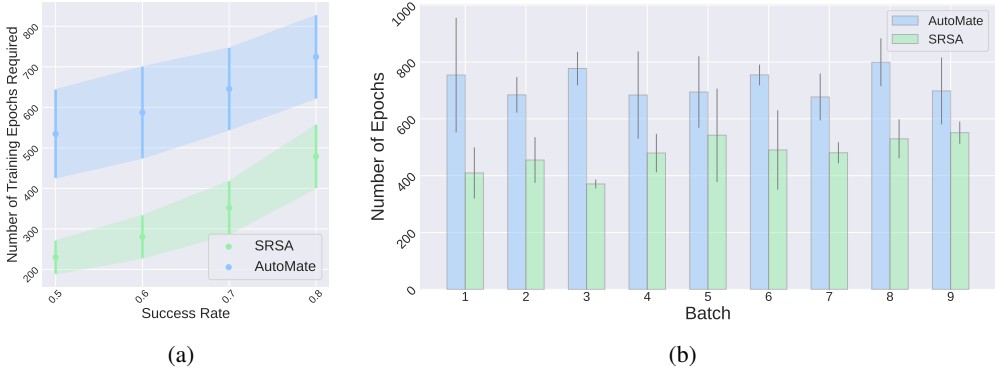

(a)  (b)

Figure 13: **(a) Overall sample efficiency**. We report the number of training epochs required to reach desired success rates (0.5, 0.6, 0.7, 0.8). We calculate the mean and standard deviation of the required training epochs over 5 runs, and report the average across 90 tasks. **(b) Sample efficiency in batches**. We sequentially introduce 9 batches of new tasks for policy learning, with each batch containing 10 new tasks. For each batch, we show the mean and standard deviation of training epochs required to reach a success rate of 0.8. SRSA consistently requires fewer training epochs.

Additionally, we compare SRSA and AutoMate based on the best checkpoint, measured by the highest rewards achieved over 5 runs for each task. In our replication of AutoMate, we achieved an average success rate of 70% across 100 assembly tasks, which is lower than the 80% reported in the original paper. This discrepancy may be due to differences in simulator versions, asset meshes, implementation details, and other factors.

On average, SRSA achieves a success rate of 79% in Fig. 14 and 73% in Fig. 15, for two cases of task ordering, respectively. SRSA demonstrates a higher success rate and better sample efficiency than the baseline AutoMate.

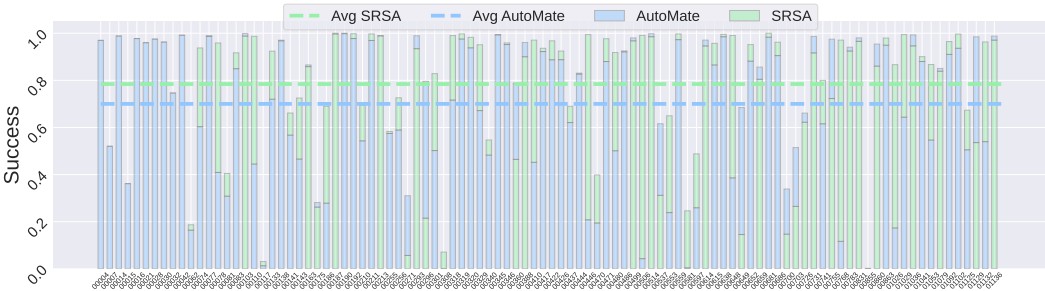

Figure 14: **Comparison of SRSA and AutoMate success rate over 100 tasks**. We replicate the specialist policy learning in the AutoMate paper over all tasks, and run SRSA with the continual-learning approach to train 90 specialist policies with the initial skill library of 10 policies. For both approaches, for each task, we select the best checkpoint among 5 runs with different random seeds. We compare the success rate on all the tasks. On average, SRSA achieves a higher success rate.

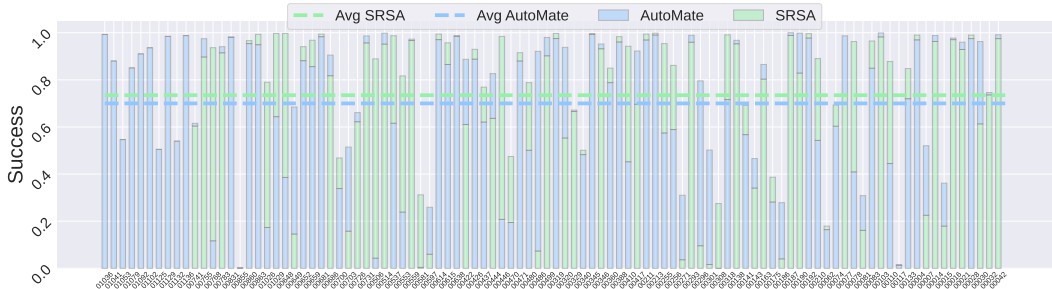

Figure 15: **Comparison of SRSA and AutoMate success rate over 100 tasks**. We replicate the specialist policy learning in the AutoMate paper over all tasks, and run SRSA with the continual-learning approach to train 90 specialist policies with the initial skill library of 10 policies. For both approaches, for each task, we select the best checkpoint among 5 runs with different random seeds. We compare the success rate on all the tasks. On average, SRSA achieves a higher success rate.

### A.5.4   ABLATION STUDY

**Implementation Details**   Fig. 16 illustrates the learning curves of different SRSA variations across 10 test tasks.

Skills retrieved based solely on geometry embeddings may face challenges during adaptation due to dynamic differences between the source and target tasks. As a result, the learning curves of SRSA-Geom tend to be less efficient and more unstable than SRSA. We further analyze this baseline in the following section.

When self-imitation learning is removed (SRSA-noSIL) from SRSA, the learning curves show increased fluctuation and higher variance across runs.

For the generalist policy, which was trained on 20 tasks from AutoMate (including tasks 01036, 01041, 01129, 01136), fine-tuning on these tasks yields strong performance since the policy was already optimized for them. However, on other test tasks, the generalist policy is not as effective for efficient policy learning compared to the skills retrieved by SRSA.

**Generalist Policy**   Fine-tuning a state-based generalist policy does not perform well because the generalist policy has limited capacity and cannot cover more than 20 training tasks.

As prior work AutoMate (Tang et al., 2024) has shown, the training success rate of a state-based generalist policy decreases significantly when the number of training tasks exceeds 20, given a fixed policy architecture of RNN and MLP. We believe that this may be because each task requires precise control across distinct geometric features, and a single policy cannot capture the strategies for all these challenging tasks.

While "increasing model capacity" or moving toward a "large data and large model" regime might help mitigate this problem, it might introduce other challenges. Simply scaling model capacity could result in a generalist policy that works well in-domain but operates more like a "switching circuit," effectively storing task-specific strategies without generalizing to out-of-domain tasks. This approach is suboptimal as it prioritizes in-domain performance at the expense of out-of-distribution generalization. Thus, we do not want to increase the model capacity indefinitely. Instead, we may need a more advanced architecture (e.g., diffusion policy) or model-based RL approach with planning to better handle diverse tasks.

That said, several open questions remain for the state-based generalist policy: How do you design policy architectures capable of high-precision control across many tasks? How do you train the generalist policy efficiently on many assembly tasks considering possible gradient conflicts? How many training tasks are needed to achieve strong out-of-distribution generalization performance on new assembly tasks?

Fine-tuning a vision-based generalist policy presents more challenges, such as effectively learning a generalist policy across multiple prior tasks with high-dimensional vision observations, fine-tuning on new tasks without forgetting prior ones, and addressing continual learning scenarios, including whether to fine-tune the original generalist policy or one already fine-tuned on other tasks. We made

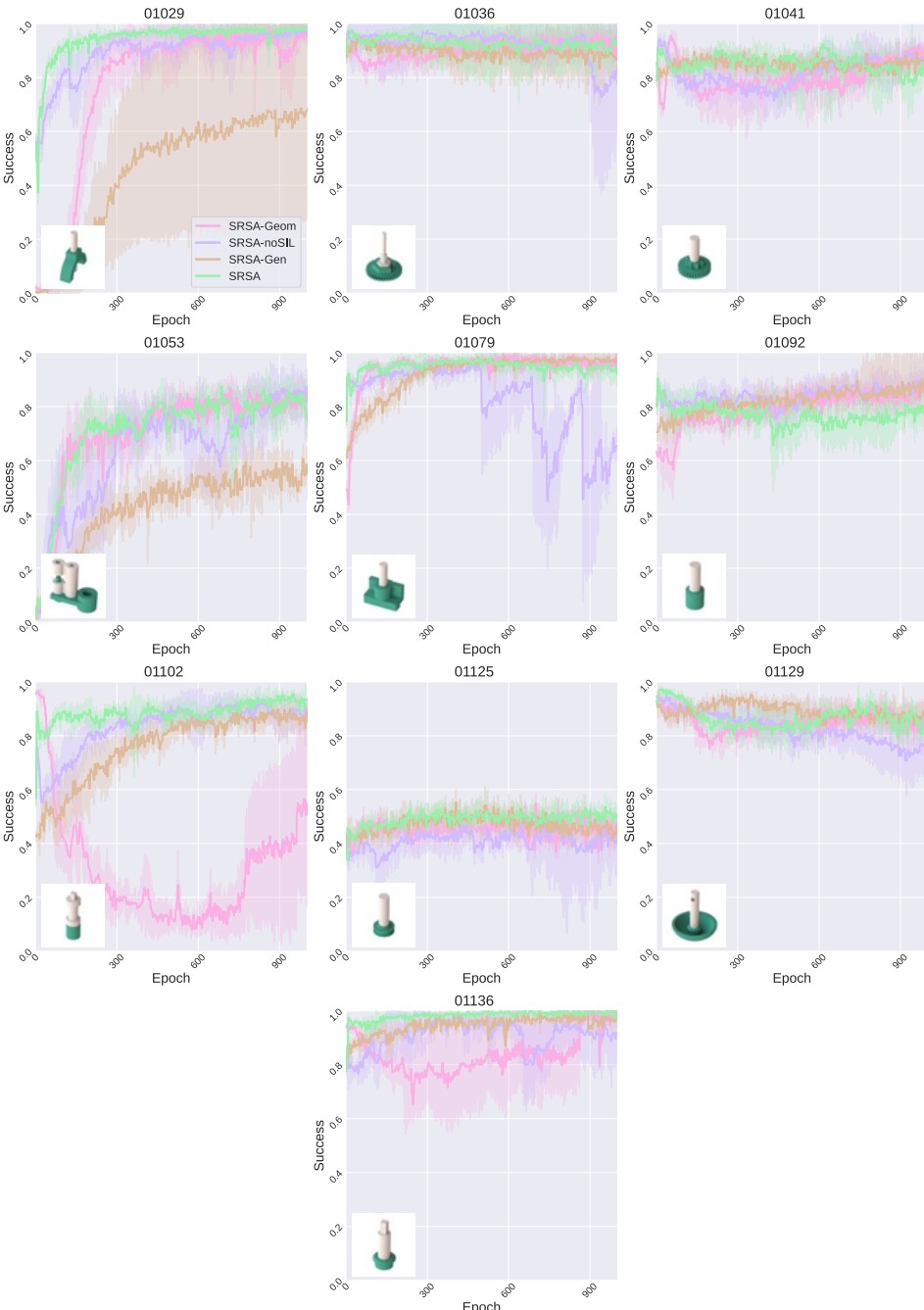

Figure 16: **Comparison for variants of SRSA with different ablated components**. For each method, we have 5 runs with different random seeds. The learning curves show mean and standard deviation of success rate over these runs.

an initial attempt to train a vision-based generalist policy with PPO and fine-tune it. Given 90 prior tasks, it can only reach around 10% average success rate. We expect such a generalist policy would perform no better than random initialization when fine-tuned for new tasks. Vision-based RL for generalist policy on assembly tasks is a relevantly new topic, and the development of such policies lies beyond the scope of SRSA. We leave this direction for future research.

A.6 COMPARISON WITH GEOMETRY-BASED RETRIEVAL

During adaptation, the final performance of SRSA-geom looks close to SRSA in some cases (see Fig. 16). However, it is statistically worse than SRSA, especially when there is a smaller number of training epochs. To provide a more comprehensive evaluation, we run SRSA-geom and SRSA across additional target tasks with three random seeds. The table below summarizes statistics of success rate at different numbers of training epochs, showing that SRSA consistently achieves higher success rates with lower variance. In industrial settings, a 2–9% difference in success rate can be highly substantial.

| Success rate (%) | Test task set 1 | | Test task set 2 | |
|---|---|---|---|---|
| | Epoch 500 | Epoch 1000 | Epoch 500 | Epoch 1000 |
| SRSA-geom | 73.6 (± 6.9) | 81.0 (±7.7) | 67.7 (±7.1) | 71.4 (±8.1) |
| SRSA | **81.4 (±4.7)** | **82.6 (±4.8)** | **76.2 (±3.0)** | **77.6 (±3.5)** |

Geometry-based retrieval alone is not always sufficient. When tasks share similar geometry but have different dynamics, SRSA-geom struggles to transfer as effectively as SRSA. For example, for the target task 01092, SRSA-geom retrieves source task 00686, achieving a transfer success rate of only 61.1%, whereas SRSA retrieves task 00213 with a higher success rate of 76.7%. Although the overall shapes of 01092 and 00686 are similar (see below), the lower part of the plug in task 01092 is thinner than the upper part, and there is only a short distance to insert this lower part into the socket. These features closely resemble task 00213, i.e., a narrow plug to be inserted a short distance to accomplish assembly. These shared physical characteristics and similar task-solving strategies make 00213 better suited for transfer. In assembly tasks, the dynamics of the contact region are often more critical than overall geometry for task success. Therefore, source task 00213 works better than 00686 when transferring to the target task 01092.

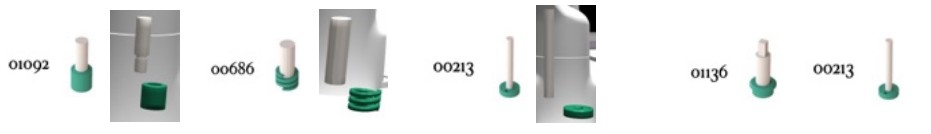

(a) Assembly tasks in the first example          (b) Assembly tasks in the second example

Additionally, we examine assembly tasks with identical geometry but differing physical parameters. For instance, consider the target task 01136 with a friction value of 10.0. One source task has the same geometry as 01136 but a significantly lower friction value of 0.5. SRSA-geom selects this source task due to its geometric similarity; however, the corresponding source policy achieves only 88.9% transfer success on the target task due to the friction mismatch (compared to a 99.3% success rate on its original source task). In contrast, SRSA selects the source task 00213, which not only shares geometric similarity but also has a friction value closer to that of the target task. As a result, SRSA retrieved policy achieve a higher zero-shot transfer success rate of 93.2%.

A.7 ANALYSIS OF SOURCE POLICY SUCCESS AS INPUT FOR RETRIEVAL

The success rate of the source policy on the source task is meaningful information to represent the source policy. To see whether it is practically beneficial for retrieval, we modify our approach. We simply concatenate the source success rate information with the task features of the source and target tasks. We train the transfer success predictor F with these features as inputs.

We consider three random splits between the prior task set (90 tasks) and test task set (10 tasks). For each split, we train F on the prior task set over three random seeds. For each seed, we test the trained function F on the test task set for retrieval. We report the mean transfer success rate of the retrieved skills on 10 test tasks, with the standard deviation reported over three seeds. Empirically, the source success rate as input to F only slightly improves the retrieval results.

| Average transfer success (%) | Test task set 1 | Test task set 2 | Test task set 3 |
|---|---|---|---|
| SRSA | 62.7 (±5.7) | 53.7 (±5.5) | **44.9 (±2.4)** |
| SRSA+source success rate | **66.7 (±0.3)** | **53.7(±2.6)** | 43.7 (±3.7) |

## A.8 Analysis of Out-of-Distribution Test Tasks

For out-of-distribution (OOD) tasks where no skill transfers zero-shot, SRSA may indeed struggle, and the initialization from a retrieved skill might not help much. To tackle this, it is essential to build a skill library which is as diverse as possible. When the target task falls outside the current library's distribution, we can use SRSA's continual learning approach (section 4.3 & 5.4) to expand the library with new tasks. By building a larger, more varied skill library, we increase the likelihood that this target task will align better with tasks in the skill library.

We run experiments for target tasks with IDs 00004, 00015, 00016, 00028, 00030. These tasks suffer from low transfer success rate given a small skill library with only 10 prior tasks. However, when we have a larger and larger skill library, the retrieved skill has a higher transfer success rate on the target task.

| Transfer success rate (%) | 00004 | 00015 | 00016 | 00028 | 00030 |
|---|---|---|---|---|---|
| 10-task library | 15.9 | 6.9 | 0.2 | 12.2 | 39.1 |
| 50-task library | 12.7 | 8.4 | 0.3 | **27.5** | 49.4 |
| 90-task library | **24.2** | **28.4** | **19.3** | 18.1 | **82.6** |

As demonstrated, continual learning to expand the skill library is a promising step; however, generalizing to OOD tasks is a longstanding challenge in robotics, and it is still an open question how to optimally construct the curriculum that governs the expansion of the skill library.

## A.9 Analysis of Other Metrics for Retrieval

We acknowledge that zero-shot transfer success rate may not be a perfect proxy for retrieval. We can consider several other possible metrics for retrieval: (1) Ground-truth success rate after adaptation (2) Predicted success rate after adaptation (3) Predicted zero-shot success rate (i.e., SRSA) (4) Predicted zero-shot dense reward.

Option 1 is the ideal metric to identify the best skill for retrieval, as our final goal is to obtain the highest success rate on the target task after adaptation. However, it introduces a chicken-and-egg problem, as we cannot get this metric without fine-tuning all candidate policies on the target task.

Option 2 requires training a predictor for the success rate after adapting any source policy on any target task. We need the training labels of the ground-truth success rate after adaptation. Unfortunately, collecting this training data would require extensive computational resources. For each source-target pair, we need at least 20 GPU hours to finish adaptation; given a skill library of 100 tasks, 200,000 GPU hours would be required to collect training data. Furthermore, it will remain intractable as the skill library becomes larger.

Option 3 (SRSA) requires much less resources to collect training data for the predictor. We only need 20 minutes on a GPU to evaluate one source policy on a target task. It thus requires 3,000 GPU hours to collect training labels. We conduct an experiment to compare the performance of Option 1 and Option 3 on two test tasks. To collect experimental results for Option 1, for each test task, we sweep all 90 source policies in our skill library. We fine-tune each source policy with one random seed to adapt to the test task and identify the best success rate after adaptation. Below we report the success rate of Option 1 and Option 3 on two test tasks, after fine-tuning for 1500 epochs.

| Success rate after adaptation (%) | Test task 1036 | Test task 1041 |
|---|---|---|
| Option 3 (SRSA) | 95.9 | 89.1 |
| Option 1 | 98.3 | 94.0 |

Option 1 is the perfect but intractable metric for retrieval. The difference in success rate between the SRSA-retrieved skill (Option 3) and the best source skill (Option 1) is less than 5% after adaptation. Therefore, although zero-shot transfer success rate is not a perfect metric for retrieval, it is a high-quality metric for retrieval in terms of both performance and computational efficiency.

Furthermore, we consider using dense reward information to guide retrieval (Option 4). We learn to predict the accumulated reward rather than success rate on the target task when executing the source policies in a zero-shot manner; then we retrieve the source policy with the highest predicted transfer reward. In the table below, we show the performance of retrieved skills when they are applied on the target tasks.

|  | Test task set 1 | | Test task set 2 | |
|---|---|---|---|---|
|  | Transfer reward | Transfer success (%) | Transfer reward | Transfer success (%) |
| Option 3 (SRSA) | **8134** | **62.7** | 7722 | **53.7** |
| Option 4 | 7976 | 54.8 | **7935** | 32.6 |

On the AutoMate task set, Option 3 (SRSA) yields slightly better skill retrievals, especially with higher transfer success on the target task. However, success rate may not accurately reflect the expected value for tasks with dense rewards; the higher transfer success rate does not mean higher transfer reward in test task set 2. Therefore, if it is critical to prioritize the reward achieved on the target task, using the transfer-reward predictor for retrieval is a reasonable choice. Conversely, if the success rate on the target task is more critical (as in our assembly tasks), transfer success would be the preferred choice as a retrieval metric.

## A.10 ANALYSIS OF DISTANCE METRICS FOR TASK FEATURES

In SRSA method, we jointly learn features from geometry, dynamics and expert actions to represent tasks, and predict transfer success to implicitly capture other transfer-related factors from tasks. To investigate the advantage of SRSA, we compare it against baselines that uses simple distance metrics for task features to determine task retrieval. Specifically, we concatenate the features of geometry, dynamics and expert actions as the task features and apply L2 distance, L1 distance, and negative cosine similarity between the vectors as the distance metrics for retrieval. We consider three different ways to split the prior task set (90 tasks) and test task set (10 tasks). For each test task, we retrieve the source task with the closest task feature to the target task. The table below shows that SRSA outperforms the baselines on all test task sets.

| Transfer success rate (%) | L2 distance | L1 distance | Cosine similarity | SRSA |
|---|---|---|---|---|
| Test task set 1 | 51.6 | 50.8 | 52.6 | **62.7** |
| Test task set 2 | 47.1 | 49.0 | 46.5 | **53.7** |
| Test task set 3 | 35.3 | 35.0 | 36.1 | **44.9** |

We attribute SRSA's advantage to its transfer success predictor $F$, which capture additional information relevant to policy transfer. By explicitly learning to predict transfer success, $F$ provides a more effective metric for selecting source tasks with higher zero-shot transfer success."

## A.11 ABLATION STUDY ON POLICY INITIALIZATION AND SELF-IMITATION LEARNING

For policy learning, AutoMate uses PPO from random policy initialization, and SRSA uses PPO with self-imitation learning (SIL) after initialization with the retrieved skill. Thus, the main difference between SRSA and AutoMate lies in (1) strong initialization from retrieval and (2) SIL.

In Sec. 6, we compared SRSA and SRSA-noSIL to show the effect of SIL. Below, we additionally compare with SRSA with random initialization (SRSA-noRetr) to show the effect of initialization from retrieval. Comparing AutoMate with SRSA-noRetr, we see the difference between PPO and PPO+SIL when learning a policy from scratch. Both approaches started from poor performance, but SIL has greater learning efficiency and stability. Comparing SRSA-noRetr and SRSA, we see the difference between random initialization and initialization from retrieval. Policy retrieval provides a good start with a reasonable success rate. As a result, SRSA more efficiently reaches higher performance on the target task.

## A.12 TOP-$k$ RETRIEVAL SELECTION IN SRSA

Although the transfer success predictor $F$ in SRSA effectively guides the retrieval of relevant skills, its predictions may not always be perfectly accurate. To mitigate this issue, we retrieve the top-

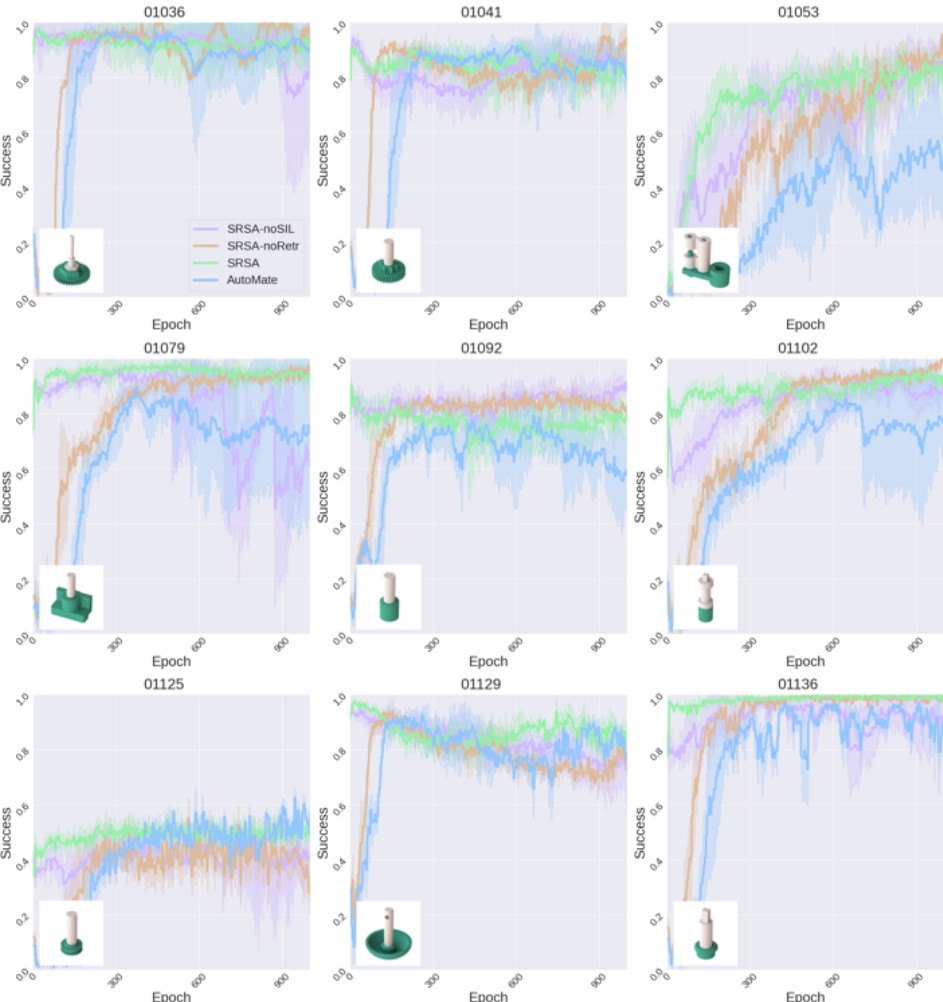

Figure 17: **Comparison for variants of SRSA with different ablated components**. For each method, we have 5 runs with different random seeds. The learning curves show mean and standard deviation of success rate over these runs.

$k$ skills ranked by the predictor $F$. With these $k$ candidates, we evaluate their zero-shot transfer success on the target task by running each candidate for 100 episodes. We then select the best candidate with the highest average reward, ensuring that we identify the most relevant skill based on actual performance rather than just the predicted ranking. In Sec. 5, we set $k$ to 5.

To illustrate the impact of this selection process, we compare SRSA with SRSA-top1. Since the predictor $F$ is not always precise, the top-1 skill based on predicted transfer success may not be the best in terms of ground-truth performance. The learning curves in Fig. 18 demonstrate that SRSA and SRSA-top1 perform similarly in most cases. However, for certain tasks (e.g., 01125, 01129, 01136), SRSA benefits from selecting among the top-5 candidates and retrieves better-performing skills compared to SRSA-top1. Here SRSA-top1 works well in our setting because the pre-trained predictor $F$ is reliable given a large 90-task prior skill library. However, with a smaller skill library, $F$ may be prone to overfitting and top-$k$ selection is probably more advantageous in this case.

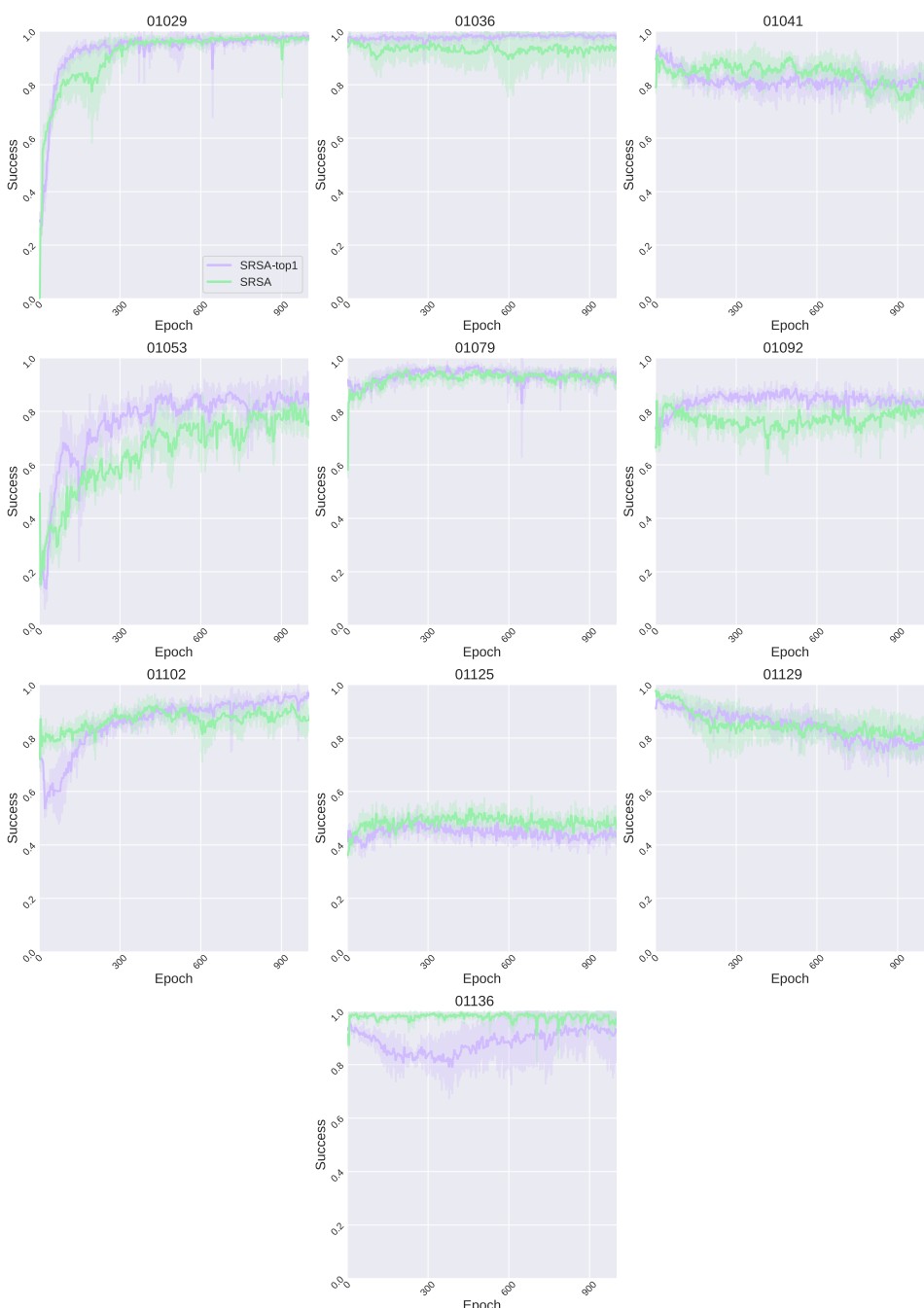

Figure 18: **Learning curves on test tasks.** The $x$-axis and $y$-axis represent training epochs (where each epoch consists of 128 environment steps over 256 parallel environments) and success rate, respectively. The solid line shows the mean success rate over 5 runs with different random seeds, and the shaded area denotes the standard deviation. SRSA takes the top-5 relevant skills based on transfer success prediction and selects one skill for policy initialization based on the ground-truth zero-shot success rate of applying the skill on the target task. SRSA-top1 directly retrieves the skill with the highest transfer success prediction.

