# OpenReview forum: "SRSA: Skill Retrieval and Adaptation for Robotic Assembly Tasks"
_ICLR.cc/2025/Conference — ICLR 2025 Spotlight_

### Official Review · Reviewer_F4Pz · 2024-10-21

**Soundness:** 3
**Presentation:** 3
**Contribution:** 2
**Rating:** 8
**Confidence:** 4

**Summary:**

This paper introduces a method for more effective learning of RL policies for robotic assembly tasks. It first uses Skill Retrieval to retrieve the most relevant policy from the skill library, a set of available policies that solve similar types of tasks. The retrieval is done based on the predicted zero-shot transfer success rate using object geometry, dynamics and expert-action features. The retrieved policy is then fine-tuned on the new task, facilitating more efficient learning when compared to learning to solve this task from scratch. The authors also showcase how their method could be used for continual learning by progressively expanding the skill library. The experiments show improved performance when compared to baselines, while the ablation studies provide valuable insights about the proposed method.

**Strengths:**

- Paper is well written and motivated.
- The proposed method is simple, elegant and easy to understand.
- Using geometry, dynamics and expert-action features for retrieval is a nice idea that incorporates different aspects of the underlying tasks.
- Experiments are conducted well and the ablation studies provide valuable insights into the method.
- Real-world experiments are appreciated.

**Weaknesses:**

- Line 217 - what does being similar with a high value mean? Maybe this sentence could be rephrased for clarity.
- Equations could be formatted better (e.g. not to overflow) and also it would be useful if they were numbered and referred to in the text when needed.
- References to the Appendix are broken as it's a separate file. This, I believe, will be addressed for the camera-ready version, just wanted to point it out.
- Using zero-shot transfer success rate as a proxy for retrieval is intuitive, but, I believe, it doesn’t accurately reflect the expected value for tasks with dense rewards. Could authors provide more details as to why using a zero-shot transfer success rate is sufficient? Would it be beneficial to use additional metrics as well?
- If the difference in expected value when applying the same policy to different tasks depends only on their difference in transition dynamics and initial state distributions that are captured by geometry, dynamics and expert-action features, why do you need to approximate the zero-shot transfer success rate at all? Wouldn't comparing these features using some distance metric be enough? Maybe authors could do some experiments on this or justify why such an approach is not appropriate.
- Relying on disassembly paths seems to restrict the method’s applicability to simple insertion tasks on a plane. Could the authors more clearly point out such a limitation and/or propose solutions to it in the paper?
- In a supervised learning setting, the loss function is usually positive. Why the objective function (line 287) is negative?
- Referring to different tasks in the experiment section using their ID number (e.g. 01125) doesn’t provide any information as to why they are hard or easy. Adding some text or image information about these tasks would benefit the understanding of the experimental results.
- In the experiments, improvements in the success rate over the baselines are provided in percentage points, which can be hard to understand given that success rate can also be expressed in percentages. For example, does a 22% increase mean a +22% or 22% increase when compared to the base? This is especially strange when seeing an 181% increase (given that the success rate is capped at 100%). Maybe the authors can clarify that, or (as I would encourage) use a different metric.
- It would be beneficial to discuss the main difference between the proposed method and the AutoMate baseline (not including the retrieval part). It would then be easier to understand the experimental results (e.g. AutoMate’s instability, line 427).
- The paper would benefit from a discussion on using visual observations, rather than object poses, and whether the proposed method would be better than fine-tuning a generalist policy that would have the capacity to generalise implicitly.
- It would also be interesting to see whether fine-tuning or learning a residual on top of the retrieved policy is more efficient. Although, this might be outside the scope of the current approach.

**Questions:**

See the Weaknesses section.

**Details Of Ethics Concerns:**

No concerns.

---

> ### Author Response · Authors · 2024-11-21
> **Response to Reviewer F4Pz [1/3]**
>
> **Question:**
> > Equations could be formatted better (e.g. not to overflow). Referring to different tasks using their ID number (e.g. 01125) doesn’t provide any information. improvements in the success rate over the baselines is hard to understand. ......
>
> **Answer:**
>
> We thank reviewer for the detailed feedback on improving clarity and presentation. In response, we have revised the text and equations in Section 4, updated numerical details in Sections 1 and 5 as well as Appendix A.5, and improved the figures in Sections 5 and 6 along with Appendix 5.4 to address these points.
>
> **Question:**
>
> >why using a zero-shot transfer success rate is sufficient? Would it be beneficial to use additional metrics as well?
>
> **Answer:**
>
> We acknowledge zero-shot transfer success may not be a perfect proxy. We consider several possible metrics for retrieval:
> (1) Ground-truth success rate after adaptation;
> (2) Predicted success rate after adaptation;
> (3) Predicted success rate in zero-shot manner (i.e. SRSA);
> (4) Predicted dense rewards in zero-shot manner.
>
> (1) Option 1 is the ideal metric to identify the best skill for retrieval, as our final goal is to obtain the highest success rate on the target task after adaptation. However, it introduces a chicken-and-egg problem, as we cannot get this metric without fine-tuning all candidate policies on the target task.
>
> (2) Option 2 requires training a predictor for the success rate after adapting any source policy on any target task. We need the training labels of the ground-truth success rate after adaptation. Unfortunately, collecting this training data would require extensive computational resources. For each source-target pair, we need at least 20 GPU hours to finish adaptation; given a skill library of 100 tasks, 200,000 GPU hours would be required to collect training data. Furthermore, it will remain intractable as the skill library becomes larger.
>
> (3) Option 3 (SRSA) requires much less resources to collect training data for the predictor. We only need 20 minutes on a GPU to evaluate one source policy on a target task. It thus requires 3,000 GPU hours to collect training labels.
> We conduct an experiment to compare the performance of Option 1 and Option 3 on two test tasks.  As for Option 1, for each test task, we sweep all 90 source policies in our skill library. We finetune each source policy to adapt to the target task and identify the best success rate after adaptation. We only afford the computational resources for two test tasks. Below we report the success rate after fine-tuning for 1500 epochs
>
> | **Success rate (%)** | **Test task 1036** | **Test task 1041** |
> |:-------------------------------------:|:------------------:|:------------------:|
> |          **Option 3 (SRSA)**          |        95.9        |        89.1        |
> |              **Option 1**             |        98.3        |        94.0        |
>
> Option 1 is the perfect but intractable metric for retrieval. The difference of success rate between the SRSA-retrieved skill (Option 3) and the best source skill (Option 1) is less than 5% after adaptation. Therefore, **although zero-shot transfer success rate is not a perfect metric for retrieval, it is a high-quality metric for retrieval in terms of both performance and computational efficiency**.
>
> (4) Inspired by the reviewer’s question, we also consider using dense reward information to guide retrieval (Option 4). We learn to predict the accumulated reward rather than success rate on the target task when executing the source policies in a zero-shot manner; then we retrieve the source policy with the highest predicted transfer reward. In the table below, we show the performance of retrieved skills when they are applied on the target tasks.
>
> |                     | **Test task set 1** |  **Test task set 1** | **Test task set 2** |  **Test task set 2** |
> |:-------------------:|:-------------------:|:--------------------:|:-------------------:|:--------------------:|
> |                     |   Transfer reward   | Transfer success (%) |   Transfer reward   | Transfer success (%) |
> | **Option 3 (SRSA)** |         8134        |         62.7         |         7722        |         53.7         |
> |     **Option 4**    |         7976        |         54.8         |         7935        |         32.6         |
>
> In the AutoMate task set, Option 3 (SRSA) yields slightly better skill retrievals, especially with higher transfer success on the target task. We acknowledge that “success rate may not accurately reflect the expected value for tasks with dense rewards”. The higher transfer success rate does not mean higher transfer reward in test task set 2. Therefore, if it is critical to prioritize the reward achieved on the target task, using the transfer-reward predictor for retrieval is a reasonable choice. Conversely, if the success rate on the target task is more critical (as in our assembly tasks), the transfer success would be the preferred choice as a retrieval metric.

---

> ### Author Response · Authors · 2024-11-21
> **Response to Reviewer F4Pz [2/3]**
>
> **Questions:**
> >why do you need to approximate the zero-shot transfer success rate at all? Wouldn't comparing these features using some distance metric be enough
>
> **Answer:**
>
> We evaluate the reviewer’s inquiry. Specifically, we concatenate the features of geometry, dynamics and expert actions as the task features, and apply some distance metrics between the vectors as the metrics for retrieval. We consider three different ways to split the prior task set (90 tasks) and test task set (10 tasks). We consider L2 distance, L1 distance, and negative cosine similarity as distance metrics. For each test task, we retrieve the source task with the closest task feature to the target task. However, the retrieval result is worse than SRSA on three different test task sets.
>
> | **Transfer success rate (%)** | **L2 distance** | **L1 distance** | **Cosine similarity** | **Transfer success prediction (SRSA)** |
> |:-----------------------------:|:---------------:|:---------------:|:---------------------:|:--------------------------------------:|
> |      **Test task set 1**      |       51.6      |       50.8      |          52.6         |                  62.7                  |
> |      **Test task set 2**      |       47.1      |       49.0      |          46.5         |                  53.7                  |
> |      **Test task set 3**      |       35.3      |       35.0      |          36.1         |                  44.9                  |
>
> As mentioned in line 527, “we jointly learn features from geometry, dynamics and expert actions to represent tasks, and predict transfer success to implicitly capture other transfer-related factors from tasks.” The SRSA learning function F aims to capture additional information for transfer success prediction. Therefore, the prediction function F provides a better metric to identify the source task with higher zero-shot transfer success.
>
>
> **Questions:**
> >Relying on disassembly paths seems to restrict the method’s applicability to simple insertion tasks on a plane.
>
> **Answer:**
>
> As mentioned in line 267, “disassembly paths can be trivially generated by employing a compliant low-level controller to lift an inserted plug from its socket and move it to a randomized pose.” We acknowledge this specific way of generating disassembly paths is ideal for top-down insertion tasks. However, **it is easy to extend our approach beyond insertion tasks on a plane**.
>
> If the socket is suspended in the air (rather than placed on a tabletop) and the insertion occurs from a random direction, we can still generate disassembly paths by pulling the plug out along the insertion axis using the low-level controller. The proposed approach remains effective with minor adjustments to the low-level controller code to accommodate this scenario.
>
> For more challenging assembly tasks (e.g., rotational or helical assembly tasks), generating disassembly paths requires additional effort. However, we can follow the procedure presented in prior work “Assemble Them All”[1] to generate disassembly paths by iterating through forces and torques along each spatial axis. **SRSA can employ this generation method to handle other assembly tasks**.
>
> If we consider general manipulation tasks instead of assembly, we might not **capture dynamics features and expert action features** from disassembly paths, but rather **from expert human demonstrations**. In recent SOTA robotics work, such as SERL [2] and $\pi_0$ [3], it has been common to employ such demonstrations for various tasks.
>
> [1]Tian, Yunsheng, et al. "Assemble them all: Physics-based planning for generalizable assembly by disassembly." ACM Transactions on Graphics (TOG) 41.6 (2022): 1-11.
>
> [2] Luo, Jianlan, et al. "Serl: A software suite for sample-efficient robotic reinforcement learning." arXiv preprint arXiv:2401.16013 (2024).
>
> [3] Black, Kevin, et al. "$\pi_0 $: A Vision-Language-Action Flow Model for General Robot Control." arXiv preprint arXiv:2410.24164 (2024).

---

> ### Author Response · Authors · 2024-11-21
> **Response to Reviewer F4Pz [3/3]**
>
> **Question:**
> > It would be beneficial to discuss the main difference between the proposed method and the AutoMate baseline (not including the retrieval part).
>
> **Answer:**
>
> As for policy learning, AutoMate is PPO from random policy initialization, and SRSA is PPO with self-imitation learning (SIL) after initialization with the retrieved skill. Thus, the main difference between SRSA and AutoMate lies in **(1) strong initialization from retrieval** and **(2) SIL**. In section 6, we compared SRSA and SRSA-noSIL to show the effect of SIL. **In Appendix A.11**, we additionally compare to SRSA with random initialization (SRSA-noRetr) to show the effect of initialization from retrieval.
>
> **(1)** Comparing SRSA-noRetr and SRSA, we see the difference between random initialization and initialization from retrieval. Policy retrieval provides a good start with a reasonable success rate. As a result, SRSA more efficiently reaches higher performance on the target task.
>
> **(2)** Comparing AutoMate with SRSA-noRetr, we see the difference between PPO and PPO+SIL when learning a policy from scratch. Both approaches started from poor performance, but SIL has greater learning efficiency and stability.
>
> **Question:**
> > The paper would benefit from a discussion on using visual observations, rather than object poses, and whether the proposed method would be better than fine-tuning a generalist policy that would have the capacity to generalise implicitly.
>
> **Answer:**
>
> In Appendix A 4.4, and A 5.4, we add the discussion on these two aspects **(1) visual observation** and **(2) generalist policy**.
>
> **(1)** **Using visual observations or object pose is orthogonal to our proposed method** (i.e., SRSA is independent of the observation modality). The idea of retrieving a relevant skill and fine-tuning the retrieved policy may also be useful if the policy has visual observation input.
>
> We follow prior work to use object poses rather than visual observations as input to the policy. Incorporating vision-based observations would introduce additional challenges for zero-shot sim-to-real transfer, as it requires a camera. In contrast, the current policy only relies on the fixed socket pose and the robot’s proprioceptive features (including the end-effector pose), making it more straightforward to execute the policy in real-world settings.
>
> **(2)** In section 6, **fine-tuning a state-based generalist policy does not perform well** because the generalist policy has limited capacity and it cannot cover more than 20 training tasks.
>
> **Fine-tuning a vision-based generalist policy presents additional challenges**, such as effectively learning a generalist policy across multiple prior tasks with high-dimensional vision observations, fine-tuning on new tasks without forgetting prior ones, and addressing continual learning scenarios, including whether to fine-tune the original generalist policy or one already fine-tuned on other tasks.
>
> To further investigate the reviewer’s inquiry, we made an initial attempt to train a vision-based generalist policy with PPO and fine-tune it. Given 90 prior tasks, it can only reach around 10% average success rate after training for two days. We expect such a generalist policy would perform no better than random initialization when fine-tuned for new tasks. Vision-based RL for generalist policy on assembly tasks is a relevantly new topic, and the development of such policies lies beyond the scope of SRSA. We leave this direction for future research.
>
> **Question:**
> >interesting to see whether fine-tuning or learning a residual on top of the retrieved policy is more efficient
>
> **Answer:**
>
> We agree that learning a residual policy can be efficient, especially when the retrieved skill is closely relevant to the target task.
>
> However, it is hard to use a residual policy approach in the context of continual/lifelong learning which is an important application of our work (as discussed in Section 4.3). SRSA offers an accelerated method to continuously acquire and compactly store new skills in a skill library. Storing compact policies is particularly valuable in robotic applications where fast inference rates are critical. In contrast, a residual policy approach may require maintaining and retrieving a hierarchy of base policies. For instance, if the retrieved policy for a new task is itself a residual policy dependent on another policy, this dependency chain can complicate storage and retrieval while also increasing the inference time.

---

> > ### Comment · Reviewer_F4Pz · 2024-11-22
> > **Response to the rebuttal**
> >
> > Thank you for your in-depth answers and additional experiments, they are really appreciated. New experiments provide valuable insights and are convincing.
> >
> > Regarding using visual observations, I am not sure if I fully agree with your statement: "The idea of retrieving a relevant skill and fine-tuning the retrieved policy may also be useful if the policy has visual observation input."
> >
> > For visual observations, you would have to somehow account for the similarity in observations and how your vision encoder processes them (maybe additional terms in the retrieval?). In general, there is no guarantee that the vision encoder would benefit from pre-training on a similar skill, as the extracted features might differ significantly. It might help, but it might actually hinder the performance. I understand that it is outside the scope of this work, but it might be useful to discuss these challenges.
> >
> > Regarding the use of generalist policy, you say: "fine-tuning a state-based generalist policy does not perform well because the generalist policy has limited capacity and it cannot cover more than 20 training tasks." Does this mean that if you just increased the model capacity, this problem would be solved? And in the large data and large model regime it would be a preferred approach?

---

> > > ### Author Response · Authors · 2024-11-23
> > > **Response to Reviewer F4Pz [1/2]**
> > >
> > > Thank you, Reviewer F4PZ, for your prompt and insightful feedback. We are glad that our additional experiments and responses addressed most of your questions. Below, we further elaborate on the two points you raised:
> > >
> > > **Question:**
> > > >For visual observations, you would have to somehow account for the similarity in observations and how your vision encoder processes them (maybe additional terms in the retrieval?). In general, there is no guarantee that the vision encoder would benefit from pre-training on a similar skill, as the extracted features might differ significantly
> > >
> > > **Answer:**
> > >
> > > We think the high-level idea of retrieving a relevant skill and fine-tuning the retrieved policy remains applicable in scenarios involving vision-based policies. **The geometry features derived from point clouds in our task representation can partially capture visual similarities between tasks**. This enables the retrieval of source tasks that are visually similar to the target task to some degree.
> > >
> > > At the same time, **we agree with your insightful observation that SRSA may require modifications to better support vision-based policies**. As you noted, there is no guarantee that the retrieved source and target tasks are visually similar enough and the features extracted by the vision encoder in policy might differ significantly on source and target tasks. This could pose challenges for fine-tuning the policy on the target task.
> > >
> > > To address this, we consider two possible directions: (1) how to perform retrieval to better account for visual similarity; (2) how to train specialist policies with visual encoders such that the current SRSA retrieval strategy is still likely to work. Below, we propose specific approaches for these two distinct directions.
> > >
> > > **(1) Enhancing retrieval by incorporating features from visual observations:** For example, integrating a Variational Autoencoder (VAE) to extract features from visual observations (as in BehaviorRetrieval [1]) and combining these with other task representations might improve the retrieval process. Additionally, learning dynamics features, such as predicting future visual observations, could implicitly encode relevant visual information in task features for retrieval.
> > >
> > > **(2) Improving the robustness of the visual encoder in policy:** Training the specialist source policy with significant data augmentation (e.g., randomizing colors, poses, backgrounds, etc.) could make the visual encoder in the source policy more robust to diverse visual observations. It is more likely to extract similar features from geometrically similar tasks. Alternatively, leveraging state-of-the-art visual foundation models (e.g., DINOv2 [2]) as visual encoders in specialist policies could further enhance generalization and robustness. These models have demonstrated strong performance in handling diverse observations and sim-to-real challenges, as shown in PoliFormer [3]. Consequently, we believe that features extracted by such visual encoders are likely to remain consistent for visual observations across geometrically similar tasks.
> > >
> > > We sincerely appreciate your thoughtful comments on this topic and have added this discussion into Appendix A.4.4 to highlight these considerations and potential future directions.
> > >
> > > [1] Du, Maximilian, et al. "Behavior retrieval: Few-shot imitation learning by querying unlabeled datasets." arXiv preprint arXiv:2304.08742 (2023).
> > >
> > > [2] Oquab, Maxime, et al. "Dinov2: Learning robust visual features without supervision." arXiv preprint arXiv:2304.07193 (2023).
> > >
> > > [3] Zeng, Kuo-Hao, et al. "PoliFormer: Scaling On-Policy RL with Transformers Results in Masterful Navigators." arXiv preprint arXiv:2406.20083 (2024).

---

> > > ### Author Response · Authors · 2024-11-23
> > > **Response to Reviewer F4Pz [2/2]**
> > >
> > > **Question:**
> > > >Does this mean that if you just increased the model capacity, this problem would be solved? And in the large data and large model regime it would be a preferred approach?
> > >
> > > **Answer:**
> > >
> > > As prior work AutoMate has shown, the training success rate of a state-based generalist policy decreases significantly when the number of training tasks exceeds 20, given a fixed policy architecture of RNN and MLP. We believe that this may be because each task requires precise control across distinct geometric features, and their single policy cannot capture the strategies for all these challenging tasks.
> > >
> > > **While “increasing model capacity” or moving toward a "large data and large model" regime might help mitigate this problem, it might introduce other challenges**. Simply scaling model capacity could result in a generalist policy that works well in-domain but operates more like a "switching circuit," effectively storing task-specific strategies without generalizing to out-of-domain tasks. This approach is suboptimal as it prioritizes in-domain performance at the expense of out-of-distribution generalization. So we do not want to increase the model capacity indefinitely. Instead, we may need a more advanced architecture (e.g. diffusion policy) or model-based RL approach with planning to better handle diverse tasks.
> > >
> > > That said, **several open questions remain for state-based generalist policy**: How to design policy architectures capable of high-precision control across many tasks? How to train the generalist policy efficiently on many assembly tasks considering possible gradient conflicts? How many training tasks are needed to achieve strong out-of-distribution generalization performance for new assembly tasks? How can we adapt to new assembly tasks without losing performance on previous ones? Addressing these questions lies beyond the scope of SRSA but represents exciting directions for future research. We are grateful for your thought-provoking questions and add this discussion in Appendix 5.4.

---

> > > > ### Comment · Reviewer_F4Pz · 2024-11-24
> > > > **Response.**
> > > >
> > > > Thank you for your answer and additional discussion.
> > > >
> > > > I'd like to thank the authors again for their responses and updates to the paper. I think all my concerns are addressed now. I believe this is a good paper and should be accepted. Therefore, I will maintain my original score of 8 (accept, good paper).

---

> ### Author Response · Authors · 2024-11-26
>
> Thank you for your thoughtful response! We truly appreciate the time you dedicated to reviewing our paper. Your feedback has greatly helped us improve this work!

---

### Official Review · Reviewer_49Jv · 2024-11-02

**Soundness:** 3
**Presentation:** 3
**Contribution:** 2
**Rating:** 6
**Confidence:** 4

**Summary:**

This paper focuses on the problem of learning to solve new assembly tasks efficiently while utilizing solutions to previously solved tasks (previously solved assembly problems). This paper achieves this using a skill retreival and adaptation approach. First, the paper assumes that there exists N set of skills where each skill can solve a particular assembly task (particular CAD model).  To solve a new task, it tries to retrieve the most similar previous CAD from the previous set of skills. This retrieval problem is solved by using a heuristic which assumes that two CADs are similar if the policy from source zero-shot solves the second task (or gives high reward in dense reward settings). This retrieval problem is used to construct a dataset to find similar tasks. This dataset is used to learn an embedding which can predict if the CAD models are similar or not (using labels from the reward heuristic).

Now for a new incoming task (or new CAD model), the best existing skill is extracted (using the learned embedding). This skill is then refined using RL. This library can then be used in a naive continual learning setting.

**Strengths:**

The paper is well written and easy to follow. Most of the proposed approach makes sense, the experimental results along with the ablations show good performance of the proposed approach.

**Weaknesses:**

*Generality:* Unfortunately, I don’t think this approach is very general. This seems a bit too tailored for assembly tasks. This  paper might fit a robotics venue much better and would probably be appreciated more in that setting. However, this is not my decision and I don’t think this should in any way count as a negative for the paper.

*Sparse reward setting:* The retrieval approach cannot work if we are in sparse reward settings, since there can be some CAD model for which no other skill transfers 0 shot. How would this approach solve this problem? This is definitely very relevant for real-world setting where dense rewards can be quite hard.

*Generalist approach (ablation):* Do you have any intuition on why the generalist approach fails? Would better feature learning or better multi-task policies help a generalist approach. Clearly, a generalist policy trained on all tasks is way more appealing. Also, curious to hear any thoughts on why the generalist policy does not perform well on all tasks. Is this because some of the assembly tasks can be very different?

*Learning embeddings for retrieval:* Given that there are less than 100 samples to learn the embedding function which uses high-dimensional input (such as point cloud embeddings). Did you find any challenges in learning this function.

*Reward heuristic:* While the simulation lemma is used as a motivation for the reward heuristic to be used in a zero-shot setting, I think there is a difference here. If two MDPs share same transition dynamics and initial state distributions then same policy will give high rewards for both. But this does not necessarily go the other way around, i.e., one policy can gives high rewards for two MDPs *does not* mean that they share similar T and S_0. I think this is an important point that should be clarified in the paper. Thus, the use of transfer should be seen as a heuristic and not a theoretical justification.

**Questions:**

see above

---

> ### Author Response · Authors · 2024-11-21
> **Response to Reviewer 49Jv [1/2]**
>
> **Question:**
> >Unfortunately, I don’t think this approach is very general. This seems a bit too tailored for assembly tasks.
>
> **Answer:**
>
> These days, many academic and industry labs are training policies for a diverse set of specialized tasks, and the idea of maintaining a skill library is becoming increasingly common. However, **how to utilize the existing policies for new tasks (rather than training from scratch) is an open and general question in robotics**. This question is relevant not just for assembly tasks, but also for general pick-and-place tasks, dexterous manipulation tasks, etc.
>
> Most robotics tasks are governed by geometry, dynamics, and behavior/action. **Our task feature learning approach is not restricted in any way to assembly tasks. Also, the idea of learning to predict zero-shot transfer success for policy transfer can be applied to other robotics tasks**. For example, for tasks that involve operating different tools, when the agent learns to deal with a new tool (e.g., pliers), employing the skill of using a tool with a similar shape and operation mechanism (e.g., scissors) could be beneficial.
>
> **We agree that the method details may need to be adjusted for other domains**. For example, the use of disassembly paths is specific for assembly (because acquiring a large number of assembly demonstrations is otherwise intractable); for other tasks, we may be able to leverage human demonstrations, which are commonly used in robot learning. Our approach can also utilize these expert demonstrations to capture dynamics and expert action information as task features.
>
> **Question:**
> > The retrieval approach cannot work if we are in sparse reward settings, since there can be some CAD model for which no other skill transfers 0 shot. How would this approach solve this problem? This is definitely very relevant for real-world setting where dense rewards can be quite hard.
>
> **Answer:**
>
> We see two main challenges here: **(1) cases where no skills transfer well in a zero-shot manner**, and **(2) the sparse-reward setting itself**.
>
> **(1)** For out-of-distribution (OOD) tasks where no skill transfers zero-shot, SRSA may indeed struggle, and the initialization from a retrieved skill might not help much. To tackle this, it’s essential to build a skill library that’s as diverse as possible. When the target task falls outside the current library’s distribution, we can use SRSA’s continual learning approach (section 4.3 & 5.4) to expand the library with new tasks. **By building a larger, more varied skill library, we increase the likelihood that this target task will align better with tasks in the skill library**.
>
> We run experiments for target tasks with IDs 00004, 00015, 00016, 00028, 00030. These tasks suffer from low transfer success rate given a small skill library with only 10 prior tasks. However, when we have a larger and larger skill library, the retrieved skill has a higher transfer success rate on the target task.
>
> | **Transfer success rate (%)** | **Test task 00004** | **Test task 00015** | **Test task 00016** | **Test task 00028** | **Test task 00030** |
> |:-----------------------------:|:-------------------:|:-------------------:|:-------------------:|:-------------------:|:-------------------:|
> |      **10-task library**      |         15.9        |         6.9         |         0.2         |         12.2        |         39.1        |
> |      **50-task library**      |         12.7        |         8.4         |         0.3         |         27.5        |         49.4        |
> |      **90-task library**      |         24.2        |         28.4        |         19.3        |         18.1        |         82.6        |
>
> As demonstrated, continual learning to expand the skill library is a promising step; however, generalizing to OOD tasks is a longstanding challenge in robotics, and it is still an open question how to optimally construct the curriculum that governs the expansion of the skill library.
>
> **(2)** As for the sparse-reward challenge, we showed the performance of fine-tuning retrieved skills when there is a sparse reward (section 5.2). SRSA obtains a higher success rate, with 139% relative improvement over the baseline approach without retrieval (e.g. AutoMate). Therefore, **under the sparse-reward scenario, techniques that do not leverage any skill library will struggle even more**.
>
> Furthermore, we can likely get a reasonable estimate of the dense reward in the real-world scenario through pose estimation. Specifically, we can estimate the time-varying pose of the plug in the robot base frame using an initial pose estimate of the plug, camera extrinsics, current forward kinematics, and the grasp pose, and we can estimate the fixed pose of the receptacle in the robot base frame using the initial pose estimate of the receptacle, camera extrinsics, and initial forward kinematics.

---

> ### Author Response · Authors · 2024-11-21
> **Response to Reviewer 49Jv [2/2]**
>
> **Question:**
> >Do you have any intuition on why the generalist approach fails? Would better feature learning or better multi-task policies help a generalist approach.
>
> **Answer:**
>
> In Section 6, We tested the strongest available generalist policy for adaptation, but it still falls short compared to SRSA. The main issue is that the generalist policy was only trained on a fixed set of 20 tasks; its performance is solid when the target task is within this set, but drops off when the task falls outside this distribution.
>
> The generalist approach faces significant challenges in scaling to a broad set of tasks. For example, prior work like AutoMate showed that the generalist policy’s success rate decreases rapidly once the number of training tasks exceeds 20. We believe that this may be because **each task requires precise control across distinct geometric features, and a single policy cannot capture the strategies for all these challenging tasks**. Improvements in task feature learning or multi-task policy architectures might help, as well as an integration of policy learning with planning.
>
> Although a generalist policy trained on assembly tasks is appealing, **achieving reliable performance across a wide variety of assembly tasks remains a significant challenge**. It faces some open questions: How can we train the generalist policy efficiently on many assembly tasks considering possible gradient conflicts? How do we design policy architectures capable of high-precision control across many tasks? How many training tasks are needed to achieve strong out-of-distribution generalization performance for new assembly tasks? How can we adapt to new assembly tasks without losing performance on previous ones? Answering these questions is out of the scope of SRSA. We leave it to future research work.
>
> **Question:**
> >Given that there are less than 100 samples, did you find any challenges in learning this function?
>
> **Answer:**
>
> Learning task features and predicting transfer success can face challenges with overfitting, particularly when the skill library contains a limited number of prior tasks. To address this, we employed **data augmentation techniques** (see Appendix A.4.1) to enhance the sample diversity for embedding learning.
> - **Geometry features**: We randomly sampled points from the mesh to generate point clouds, significantly increasing the number of samples available for geometry embedding learning beyond the number of tasks.
> - **Dynamics and expert action features**: We extracted transition sequences from 100 disassembly trajectories for each of the 100 tasks and each disassembly trajectory included a random final pose. This further helped mitigate overfitting.
>
> These techniques proved effective in our experiments. In Section 5.1, SRSA successfully retrieves relevant source tasks with high transfer success rates on unseen target tasks. The retrieval results are clearly superior to an overfitted model, which would fail to retrieve meaningful skills for unseen tasks.
>
> **Question:**
> >the use of transfer should be seen as a heuristic and not a theoretical justification.
>
> **Answer:**
>
> Thank you to the reviewer for raising this point. We revise our language in Section 4.1 and Appendix A.2 to soften our assertion, acknowledging that it is better framed as a heuristic for intuitive motivation rather than a strict theoretical justification.
>
> "According to the simulation lemma in RL theory,  the difference in expected value when applying the same policy to different tasks **partially depends on** their difference in transition dynamics and initial state distributions. If we execute a source policy $\pi_{src}$ on both the source task $T_{src}$ and the target task $T_{trg}$, the success rates $r_{src,src}$ and $r_{src,trg}$ (on $T_{src}$ and $T_{trg}$, respectively) reflect the expected value. Notably, similar success rates on these tasks indicate that their transition dynamics functions and initial state distributions **might** also be similar. Here, our success rate on the source task $r_{src,src}$ will naturally be high, because the source policy $\pi_{src}$ is already an expert policy on $T_{src}$. Thus, when the zero-shot transfer success rate $r_{src,trg}$ (i.e., applying $\pi_{src}$ directly to $T_{trg}$) is also high (e.g., similar to $r_{src,src}$), it suggests that the two tasks **might** be closely aligned in terms of their dynamics. **Therefore, we use the high transfer success rate as a heuristic indicator of similar dynamics between source and target tasks**. Details are in Appendix A.2."

---

> ### Author Response · Authors · 2024-11-25
>
> Dear Reviewer 49Jv,
>
> We sincerely appreciate the time and effort you have dedicated to reviewing our paper. To address your valuable questions, we have made the following significant updates:
>
> 1. Provided further insights on handling out-of-distribution target tasks, supported by quantitative experimental results (Appendix A.8).
> 2. Clarified the intuitive heuristic behind using zero-shot transfer success rate as an indicator for retrieval (Appendix A.2 & Section 4.1).
>
> We hope that these revisions, along with our detailed responses, comprehensively address your concerns. Please let us know if these answer your questions. If there are any remaining points that require further clarification, please feel free to let us know.
>
> Thank you once again for your insightful feedback and thoughtful review.
>
> Best regards,

---

> > ### Comment · Reviewer_49Jv · 2024-11-27
> > **Thank you for the clarifications**
> >
> > I appreciate the clarifications from the authors. I think this is a good paper. But I am still unconvinced by the generality of this approach. While I agree that many industry/academic labs are looking at skill libraries, the only skill being used in this work is "insertion". Hence, it is not a truly general multi-skill library. But this is besides the point of this paper.
> >
> > Overall, I would maintain my rating. Again, thank you to the authors for their clarifying comments.

---

> > > ### Author Response · Authors · 2024-11-27
> > >
> > > Thank you for your response! We plan to expand SRSA to a broader range of manipulation tasks to investigate its generalizability beyond the current scope, and will leave it for future work. We truly appreciate the time you dedicated to reviewing our paper. Your feedback has greatly helped us improve this work!

---

### Official Review · Reviewer_YitF · 2024-11-04

**Soundness:** 3
**Presentation:** 3
**Contribution:** 3
**Rating:** 8
**Confidence:** 4

**Summary:**

Skill retrieval and Skill Augmentation (SRSA) tackles the problem of quickly learning new tasks in task family. In the paper, the authors consider various insertion tasks from the AutoMate benchmark which share the same observation space, action space and reward function. The tasks differ in the geometries and thereby the transition dynamics and expert actions required.

The authors assume that an availability of a skill library containing expert policies for different insertion tasks. Skill retrieval is based on geometry similarity, transition dynamics and expert actions. Transition dynamics and expert actions are based on reverse trajectory generation. The similarity is predicted using a learnt function F, which is a 0-shot policy transfer success rate predictor trained on existing skill library. Given a new task, the policy with the highest 0-shot transfer prediction is used extracted and fine-tuned on the new task using Reinforcement Learning (PPO) with additional Self Imitation Learning (SIL) objective.

**Strengths:**

1. Strong simulation results
2. Insightful that we can train a 0-shot transfer predictor function using geometry, transition and expert action encodings
3. Results transfer to real robot

**Weaknesses:**

1. Assumption that that all tasks have the same observation space, action space and reward function
2. Based on Fig. 6 and Appendix Fig.16, it seems SRSA-geom is very comparable to SRSA. Hence, it seems geometry based skill retrieval might be sufficient. It's also easier as it doesn't require to collect disassembly trajectories and training a 0-shot transfer predictor function

**Questions:**

Should F use as input the success rate of the expert policy in the original task?

---

> ### Author Response · Authors · 2024-11-21
> **Response to Reviewer YitF [1/2]**
>
> **Question:**
> >Assumption that all tasks have the same observation space, action space, and reward function
>
> **Answer:**
> Our approach requires that tasks share the observation and action spaces. The assumption of a shared observation and action space is reasonable in flexible automation settings commonly found in industry. In such scenarios, a robot operating within a specific workcell typically has a fixed observation and action space, but it is expected to interact with a variety of different parts.
>
> However, we do not require the reward function to be the same across tasks.
> - For tasks in the skill library, we don’t assume anything specific about how their rewards were structured. The retrieval process doesn’t rely on reward information to identify relevant skills.
> - For the target tasks, we allow flexibility in the reward setting. In the sparse-reward experiment in Section 5.2, the target task uses a sparse reward, whereas the retrieved source skill was trained with a dense reward. In this case, we initialize the actor network on the target task using the retrieved skill, but randomly initialize the critic network for adaptation rather than copying it from the retrieved skill, so differences in reward functions on source and target tasks don’t impact adaptation.
>
> **Question:**
> >Based on Fig. 6 and Appendix Fig.16, it seems SRSA-geom is very comparable to SRSA. Hence, it seems geometry based skill retrieval might be sufficient
>
> **Answer:**
>
> During adaptation, the final performance of SRSA-geom looks close to SRSA in some cases. However, it is statistically worse than SRSA, especially when there is a smaller number of training epochs. To provide a more comprehensive evaluation, we run SRSA-geom and SRSA across additional target tasks with three random seeds. The table below summarizes statistics of success rate at different numbers of training epochs, showing that **SRSA consistently achieves higher success rates with lower variance**. In industrial settings, a 3–9% difference in success rate can be significant.
>
> |                  | **Test task set 1** | **Test task set 1** | **Test task set 2** | **Test task set 2** |
> |:----------------:|:-------------------:|:-------------------:|:-------------------:|:-------------------:|
> | **Average Success rate (standard deviation)** |      Epoch 500      |      Epoch 1000     |      Epoch 500      |      Epoch 1000     |
> |   **SRSA-geom**  |      73.6 (6.9)     |      81.0 (7.7)     |      67.7 (7.1)     |      71.4 (8.1)     |
> |     **SRSA**     |      82.8 (4.2)     |      84.7 (3.4)     |      76.2 (3.0)     |      77.6 (3.5)     |
>
> Furthermore, **geometry-based retrieval alone is not always sufficient**. When tasks share similar geometry but have different dynamics, SRSA-geom struggles to transfer as effectively as SRSA.
> - For example, for the target task 01092, SRSA-geom retrieves source task 00686, achieving a transfer success rate of only 61.1%, whereas SRSA retrieves task 00213 with a higher success rate of 76.7%. While the overall shapes of 01092 and 00686 are similar (see details in Appendix A.6), the lower part of plug in task 01092 is thinner than the upper part, and there is only a short distance to insert this lower part into the socket. These features closely resemble task 00213, i.e., a narrow plug to be inserted a short distance to accomplish assembly. These shared physical characteristics and similar task-solving strategies make 00213 better suited for transfer. In assembly tasks, the dynamics of the contact region are often more critical than overall geometry for task success. Therefore, source task 00213 works better than 00686 when transferring to the target task 01092.
> - Additionally, we examine assembly tasks with identical geometry but differing physical parameters. For instance, consider the target task 01136 with a friction value of 10.0 (see details in Appendix A.6). One source task has the same geometry as 01136 but a significantly lower friction value of 0.5. SRSA-geom selects this source task due to its geometric similarity; however, the corresponding source policy achieves only 88.9% transfer success on the target task, due to the friction mismatch (despite achieving a 99.3% success rate on its original source task). In contrast, SRSA selects the source task 00213, whose policy better aligns with the target task's dynamics, resulting in a higher transfer success rate of 93.2%

---

> ### Author Response · Authors · 2024-11-21
> **Response to Reviewer YitF [2/2]**
>
> **Question:**
> >Should F use as input the success rate of the expert policy in the original task?
>
> **Answer:**
>
> The success rate of the source policy on the source task is meaningful information to represent the source policy. To see whether it is practically beneficial for retrieval, we modify our approach. We simply concatenate this source success rate information with the task features of source and target tasks. We train the transfer success predictor F with these features as inputs.
>
> We consider three random splits between the prior task set (90 tasks) and test task set (10 tasks). For each split, we train F on the prior task set over three random seeds. For each seed, we test the trained function F on the test task set for retrieval. We report the mean transfer success rate of the retrieved skills on 10 test tasks, with the standard deviation reported over three seeds.  Empirically, the source success rate as input to F only slightly improves the retrieval results.
>
> | **Average transfer success (standard deviation)** | **Test task set 1** | **Test task set 2** | **Test task set 3** |
> |:-------------------------------------------------:|:-------------------:|:-------------------:|:-------------------:|
> |                      **SRSA**                     |      62.7 (5.7)     |      53.7 (5.5)     |      44.9 (2.4)     |
> |            **SRSA+source success rate**           |      66.7 (0.3)     |      53.7(2.6)      |      43.7 (3.7)     |

---

> ### Author Response · Authors · 2024-11-25
>
> Dear Reviewer YitF,
>
> We sincerely appreciate the time and effort you have dedicated to reviewing our paper. To address your valuable questions, we have made the following significant updates:
>
> 1. Conducted additional analysis to compare SRSA-geom and SRSA, identifying the limitations of SRSA-geom (Appendix A.6).
> 2. Performed new experiments incorporating the source success rate into task features for transfer success prediction (Appendix A.7).
>
> We hope that these revisions, along with our detailed responses, comprehensively address your concerns. Please let us know if these answer your questions. If there are any remaining points that require further clarification, please feel free to let us know.
>
> Thank you once again for your insightful feedback and thoughtful review.
>
> Best regards,

---

> > ### Comment · Reviewer_YitF · 2024-11-26
> > **Thank you for the additional results, I have increased my rating**
> >
> > Thank you for the additional results and answering my questions. These analyses have made the paper's contributions more crisp. Hence, I have increased my rating to Accept.

---

> > > ### Author Response · Authors · 2024-11-26
> > >
> > > Thank you for your thoughtful response! We truly appreciate the time you dedicated to reviewing our paper and revisiting your score. Your feedback has greatly helped us improve this work!

---

### Author Response · Authors · 2024-11-21
**General Response to Reviewers**

We sincerely thank reviewers for thorough evaluation of our paper and for thoughtful and constructive feedback, which has been super helpful in improving this work.

We greatly appreciate the positive comments highlighting that the paper is "well-motivated", "well-written" and "easy to follow" (49Jv, F4Pz). We are grateful that reviewers describe our method as "simple, elegant" and "insightful" (YitF, F4Pz). The experimental results were noted as "strong", demonstrating "good performance" and providing "valuable insight into the method" (YitF, 49Jv, F4Pz). And the importance of real-world experiments is also acknowledged (YitF, F4Pz).

According to the reviewers' questions and suggestions, we have conducted several new experiments and analyses to further strengthen the paper. Additionally, we have revised the manuscript, with all changes highlighted in blue text, summarized as follows:

- **Improved Clarity and Formatting** (F4Pz)
- **Clarification on Intuition from Theoretical Perspective** in Appendix A.2 (49Jv)
- **Comparison with Geometry-based Retrieval** in Appendix A.6 (YitF)
- **Analysis of Source Policy Success as Input for Retrieval** in Appendix A.7 (YitF)
- **Analysis of Out-of-Distribution Test Tasks** in Appendix A.8 (49Jv)
- **Analysis on Other Metrics for Retrieval** in Appendix A.9 (F4Pz)
- **Analysis on Distance Metrics for Task Features** in Appendix A.10 (F4Pz)
- **Ablation Study on Policy Initialization and Self-Imitation Learning** in Appendix A.11 (F4Pz)

Below, we respond to each reviewer separately to address questions in detail.

---

### Meta-Review · Area_Chair_tttQ · 2024-12-16

**Metareview:**

The paper proposes SRSA, an approach that improves the efficiency with which robots are able to learn novel assembly tasks by leveraging an existing skill library. Given a new task, SRSA identifies the known skill that is predicted to yield the highest zero-shot success rate based upon the object geometry, dynamics, and expert-action features. SRSA then fine-tunes the retrieved skill on the new task. The paper describes how SRSA can be used in a continual learning setting, whereby the skill library is continuously expanded. Experiments demonstrate that SRSA outperforms a contemporary baseline in terms of learning efficiency and task success.

The ability for robots to learn to perform new task by adapting existing policies as opposed to learning from scratch is of practical importance and is a problem of interest to many in the robot learning community. The means by which SRSA performs retrieval using a combination of object geometry, dynamics and expert-action features is compelling, while the overall framework has the benefit of being straightforward. Despite its relative simplicity, as the reviewers point out, the simulation-based experiments reveal strong benefits in terms of performance over the baseline method. The reviewers further appreciated the results that demonstrate the ability to transfer SRSA to real robots. Additionally, several reviewers noted that the paper is well written.

Some of the reviewers initially raised concerns about the generality of the method beyond assembly tasks, some of the assumptions on which the method relies, and the sufficiency of the experimental evaluation (specifically the comparison to the generalist baseline and the SRSA-geom ablation). The authors made a concerted effort to address these concerns, which included the addition of new experimental results. This effort together with the discussion between the authors and reviewers helped to clarify the significance of the paper's contributions. After discussing the paper with the reviewers, the AC acknowledges that the paper's immediate impact may be limited by the relatively narrow scope of the assembly domain. However, the AC agrees with several of the reviewers that the core ideas would be beneficial to the broader community.

**Additional Comments On Reviewer Discussion:**

There was a healthy amount of discussion between the authors and reviewers. This helped to clarify the significance of the paper's contributions as indicated by the post-rebuttal comments provided by several reviewers. Additionally, the AC discussed concerns about the generality of the method raised by one reviewer. Other reviewers acknowledged thee focus on robot assembly, but felt appreciated the paper's contributions and potential nonetheless.

---

### Decision · Program_Chairs · 2025-01-22

Accept (Spotlight)